# LLM-Mediated Guidance of MARL Systems

## Abstract

In complex multi-agent environments, achieving efficient learning and desirable behaviours is a significant challenge for Multi-Agent Reinforcement Learning (MARL) systems. This work explores the potential of combining MARL with Large Language Model (LLM)-mediated interventions to guide agents toward more desirable behaviours. Specifically, we investigate how LLMs can be used to interpret and facilitate interventions that shape the learning trajectories of multiple agents. We experimented with two types of interventions, referred to as controllers: a Natural Language (NL) Controller and a Rule-Based (RB) Controller. The NL Controller, which uses an LLM to simulate human-like interventions, showed a stronger impact than the RB Controller. Our findings indicate that agents particularly benefit from early interventions, leading to more efficient training and higher performance. Both intervention types outperform the baseline without interventions, highlighting the potential of LLM-mediated guidance to accelerate training and enhance MARL performance in challenging environments.

## 1 Introduction

Cooperative MARL research has developed techniques to effectively optimize collective return in simulated environments (Rashid et al., 2020; Yuan et al., 2023; Albrecht et al., 2024). This enables the deployment of multi-agent systems (MAS) that can efficiently solve complex tasks, particularly in tasks that factorize into parallel subtasks and/or take place in the physical world (e.g., robotics) and can benefit from spatially-scattered agents (Calvaresi et al., 2021). However, what if the reward function is misspecified? This can happen because the reward is difficult to define in a way that avoids reward hacking (Skalse et al., 2022). Alternatively, what if the test time environment or system goals change slightly? We would like a user to be able to steer a MARL system towards more desirable behaviour (human-in-the-loop). These are all key challenges that arise in real-world domains. In addition, we do not want to assume the user is a MARL expert. Ideally, the user could steer the system in an intuitive and simple way. Therefore, we consider steering a MAS using natural language. The user issues high-level strategies that an LLM then translates into actions to communicate with the MAS. While examples of humans intervening and controlling static programs/interfaces via LLMs are pervasive (Hong et al., 2023), we know of fewer examples controlling single-agent *learning* systems and no examples controlling MA learning systems.

Integrating LLMs with RL presents exciting opportunities for enhancing agent performance, particularly in complex MA environments. Instruction-aligned models with advanced reasoning and planning capabilities are well-suited for this task. Prompted correctly, these models provide real-time, context-aware strategies, guiding agents through challenges where traditional RL methods struggle, especially in environments with large action/observation spaces or sparse rewards, particularly during early training. We envision a future where LLM-RL combinations can manage increasingly dynamic environments, with LLMs handling complex interactions and dynamically changing observation and action spaces. Our research explores this potential in MARL. We allow users to quickly 'fine-tune' a base MARL system by guiding the agents using free-form natural language or rule-based interventions in the training process. This adaptation helps the system align more closely with the user's bespoke task requirements, ensuring that agents develop behaviours tailored to the

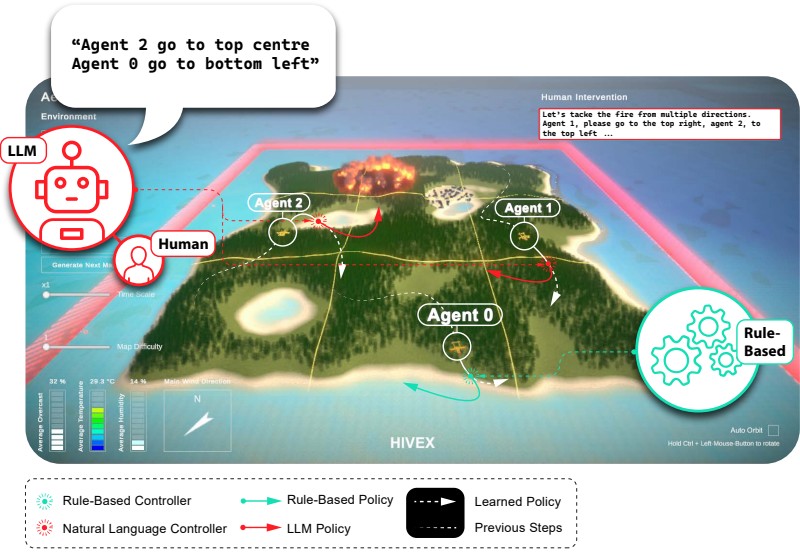

Figure 1: The Aerial Wildfire Suppression environment includes two types of controllers: Natural Language-based and Rule-Based. Controller interventions are passed to the LLM-Mediator, temporarily providing actions and overwriting the agents' learned policy actions.

challenges of the environment. We have specifically chosen the Aerial Wildfire Suppression (AWS) environment from the HIVEX suite[1], as it offers a relevant and intricate problem to solve.

The AWS environment presents dynamic and high-stakes cooperative scenarios, where the unpredictability of wildfire spread creates an evolving challenge. Factors such as wind direction, humidity, terrain slope, and temperature—hidden from the agents—add layers of complexity. Solving this environment requires seamless collaboration among agents, where strategic coordination is essential to containing fires. With AWS, users engage in a problem simulating real-world wildfire management. The combination of a physically and visually rich simulation, open-ended scenarios and environmental conditions makes AWS a demanding environment and a great challenge.

In this work, we test whether combining current MARL and LLM techniques can allow users to steer and guide a MARL system towards more desirable behaviour in the challenging AWS environment. We consider two users: the simple Rule-Based (RB) Controller and a more sophisticated Natural Language (NL) Controller. The NL Controller simulates how humans might interact with the MAS, i.e., in free-form natural language. We compare these against our baseline, a setup with no test-time interventions. We summarize our core contributions as follows:

- **Rule-Based and Natural Language Controller Generated Interventions**: We implement a novel system where rule-based and natural language-based interventions demonstrate the ability to enhance decision-making and coordination in dynamic settings like AWS.

- **Adaptive and Dynamic Guidance**: Our approach moves beyond static curriculum-based methods, providing real-time, adaptive interventions that respond to the evolving states of agents and environments, improving both long-term strategy and immediate decision-making.

- **AWS Environment**: We apply our method to the HIVEX AWS environment, simulating coordinated aerial wildfire suppression, showcasing the effectiveness of LLM-mediated interventions in managing complex and dynamic tasks in a MA environment.

---

[1]Environment: https://anonymous.4open.science/r/hivex-environments-7D23
Training Code: https://anonymous.4open.science/r/llm_mediated_guidance-0B22
Results: https://anonymous.4open.science/r/hivex-results-6438

- **Accelerated Learning and Improved Coordination**: Our results demonstrate that interventions, especially during early training, accelerate learning to reach expert-level performance more efficiently.

## 2 RELATED WORK

Integrating LLMs into RL has become pivotal for enhancing agent performance in complex environments. Advanced LLMs, specifically, their instruction fine-tuned versions, have demonstrated significant capabilities in providing high-level guidance, common-sense reasoning, and strategic planning, thereby possibly improving RL agents' adaptability and generalization (Bubeck et al., 2023). Recent works, such as those by Wang et al. (2023) and Chiang & Lee (2023), have shown that LLMs can assist RL agents by mediating natural language instructions and guiding behaviours, especially in environments where traditional reward signals are sparse or ineffective (Kajić et al., 2020). However, these studies primarily focus on single-agent scenarios or environments with relatively straightforward dynamics. In contrast, our work emphasizes MA environments with complex, interdependent dynamics, demonstrating that LLM-driven interventions can significantly accelerate learning in such settings.

Historically, human-in-the-loop RL involved human feedback in guiding the learning process (Kamalaruban et al., 2019). LLMs have emerged as scalable, real-time alternatives, providing domain-specific knowledge and policy suggestions to correct suboptimal behaviours (Chiang & Lee, 2023). While previous research by Narvekar et al. (2020) explored dynamic curriculum approaches, where models generate instructions that change based on the agent's progress, our approach leverages LLMs not for curriculum generation but for real-time human and LLM-based interventions specifically designed to address the challenges of coordinating multiple agents. This key distinction significantly impacts the effectiveness of the learning process in more complex environments. LLMs also address challenges in long-term planning and common-sense reasoning (Hao et al., 2023) by offering early and intermediate guidance that traditional RL methods often lack. Previous studies in robotics have similarly leveraged LLMs as high-level strategic planners, enabling more effective decision-making in tasks that require long-term coordination and planning (Tang et al., 2023; Ahn et al., 2022). While these works illustrate the potential of LLMs in improving decision-making in tasks requiring extended sequences of actions, our work expands this concept by integrating LLM-driven interventions at critical points in the learning process, specifically in MA scenarios where coordinated action over long horizons is crucial.

In MA systems, LLMs show promise in improving coordination and strategic planning. Traditional MARL approaches, like MADDPG and QMIX, face limitations due to the complexity of joint action spaces and sparse rewards (Lowe et al., 2017; Rashid et al., 2018). Other work specifies a mediator to steer an MA system towards a desirable equilibrium without incorporating any LLM (Zhang et al., 2024). While recent works, such as Kwon et al. (2023), have demonstrated that a global reward can control an MA system with a single intervention at the beginning—showing how to cheaply design a reward model in natural language using an LLM—these approaches do not fully address the dynamic nature of MA environments where frequent adaptations are necessary (Wang et al., 2024). Our research builds on these insights by demonstrating that periodic LLM interventions significantly enhance cooperation and learning efficiency, especially in dynamic and unpredictable environments such as AWS. This adaptive intervention strategy addresses the shortcomings of static coordination approaches by providing real-time guidance that aligns with the evolving state of the environment and agent interactions.

LLM interventions offer adaptive guidance that complements traditional policy shaping (Griffith et al., 2013), evolving with the learning process. Our method does not fit neatly into Open-loop or Closed-loop categories (Sun et al., 2024), as it temporarily replaces RL agent actions with LLM-guided interventions in both NL and RB setups. Unlike prior work using LLMs for agent communication and collaboration, our approach uniquely employs a central LLM to craft high-level strategies for coordinating multiple agents. This aligns with open research directions, specifically "language-enabled Human-in/on-the-Loop Frameworks" (Sun et al., 2024), by mimicking human-in-the-loop strategies. In contrast to Wang et al. (2023), which focuses on building agent capabilities, we emphasize centralized LLM-driven strategy development. Whether through strategic foresight or moment-to-moment decision-making, our approach adapts to dynamic environments. Assuming

we only compare the inference cost of our LLM-Mediator module, we gain an advantage as long as its cost is lower than the total inference cost of the agent over deployment.

# 3 THE AERIAL WILDFIRE SUPPRESSION ENVIRONMENT

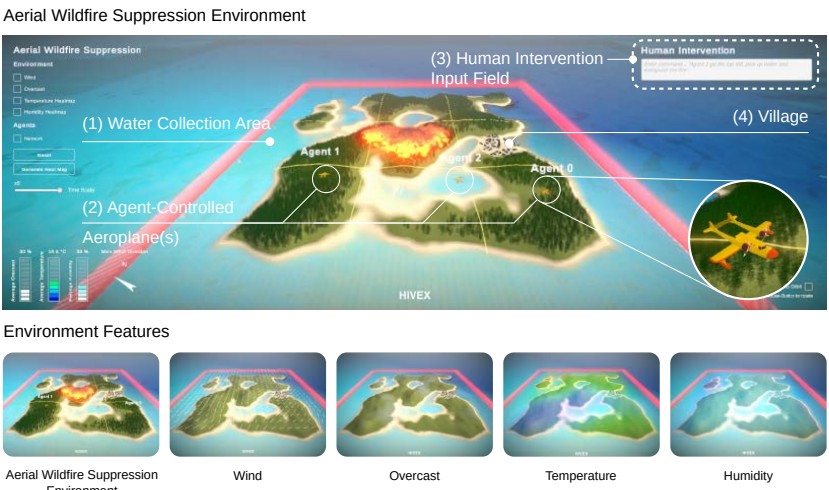

Figure 2: AWS Environment: (1) Water Collection Area, (2) Agent-controlled Wildfire Suppression Aeroplanes, (3) Human Natural Language Controller Input Field, (4) Village. Environment Features: Wind, overcast, temperature and humidity map sample.

The AWS environment presents a rich and challenging scenario for AI agents, far exceeding the simplicity of traditional grid-based worlds. Unlike grid worlds, which offer limited spatial complexity, this environment presents a three-dimensional, continuous, and dynamic landscape where agents must adapt to fire spread patterns that are difficult to predict. AWS is built in Unity (Juliani et al., 2020), a game development engine, offering a saturated, semi-realistic-looking visual component compared to Atari-like environments (Mnih et al., 2013), providing a more complex and high-dimensional observation space with both feature vector and visual data. This diversity of input, combined with the need for real-time decision-making and collaboration, makes it a robust and challenging platform for testing advanced AI strategies in complex, non-deterministic scenarios.

The AWS environment simulates a complex scenario where agents must manage and mitigate the spread of wildfires. This environment is designed to challenge agents with complex decision-making tasks, requiring both individual action and coordinated teamwork. The main focus is on reducing fire spread, protecting key assets, the village, and navigating a large, bounded terrain. The agent's primary objective is to minimize the fire's burning duration by extinguishing as many burning trees as possible and preparing unburned areas to prevent further spread. Agents can either extinguish burning trees or redirect the fire's path by preparing/wetting the surrounding forest area.

The environment includes three agents, each with a feature vector ($\mathbb{R}^8$) and visual-observation space ($42 \times 42$ RGB grid). Feature vector observations include agent 2-d position, direction, a binary indicator of whether the agent is holding water, position of the nearest tree, and the nearest tree's state, burning or not burning. The agents move at a constant velocity with actions to steer left, right, and drop water if held. They operate within a bounded area on an island. A negative reward is given if the agent crosses the environment's boundary. Water surrounds the island; steering the aeroplane toward and collecting it produces a positive reward. Agents earn positive rewards for extinguishing or preparing forest areas to slow fire spread and for extinguishing the wildfire completely. Detailed environment specifications A.4, detailed task list, reward breakdown and calculations can be found in the Appendix in Reward Description and Calculation A.5.

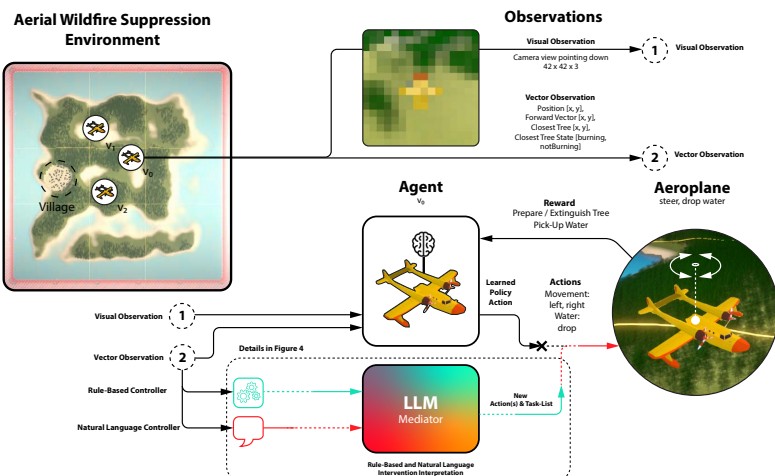

Figure 3: AWS Process Diagram: The default setup consists of three agents controlling individual aeroplanes. Each agent receives both feature vector and visual observations. Agents' actions include steering left, right, or releasing water. Rewards are given for extinguishing burning trees; smaller rewards are given for wetting living trees and picking up water. A negative reward is given for crossing the environment boundary. The LLM-Mediator interprets RB and NL Controller interventions, assigning tasks to any agent for the next 300 steps and overwriting its policy actions.

# 4 INTERVENTION CONTROLLERS AND LLM-MEDIATOR

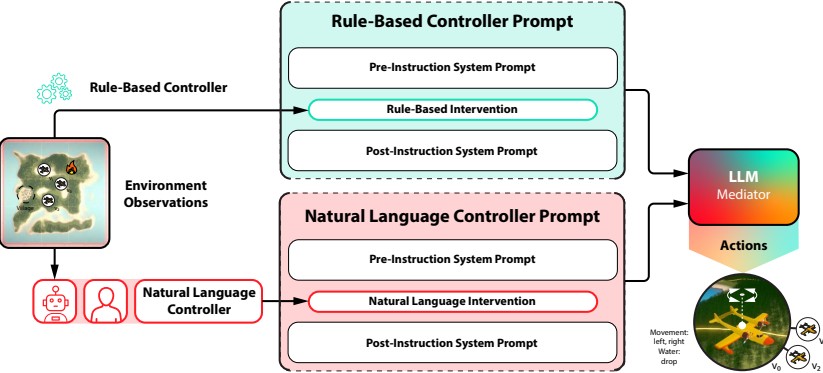

Figure 4: Overview of simplified RB and NL Controller intervention prompts sent to the LLM-Mediator, overwriting the agents' learned policy actions.

Our system supports interventions from two types of controllers: the Rule-Based (RB) and Natural Language (NL) Controller, which differ in their level of sophistication for generating interventions. The RB Controller uses predefined rules and a prompt template, producing rudimentary agent instructions. In contrast, the NL Controller communicates in free-form natural language, mimicking human behaviour. This allows it to generate more complex strategies and contextually relevant guidance. The LLM-Mediator processes both types of interventions, translating them and temporarily overwriting the agents' learned policy actions, guiding them to complete specific tasks (Figure 5). This framework enables adaptive guidance and control in dynamic environments (Figure 4).

## 4.1 RULE-BASED (RB) CONTROLLER

The RB Controller uses a prompt template that includes a subset of the agents' feature vector observations. This subset contains the agent's position and detected fire locations, which are pre-processed to natural language and integrated into the prompt template before being passed to the

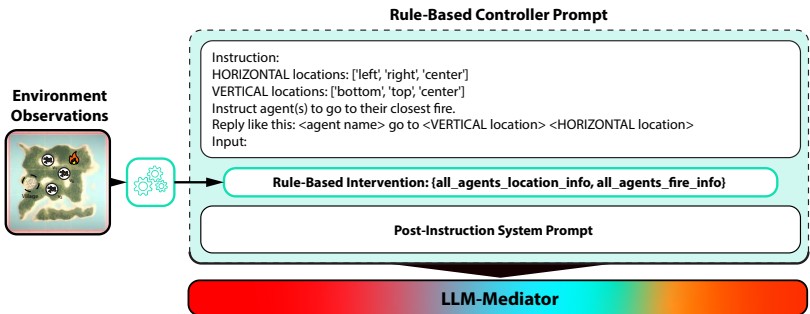

Figure 5: Abbreviated Rule-Based Controller intervention prompt template. A complete version can be found in the Appendix 12.

LLM-Mediator. The RB Controller's directive is to "*Instruct agent(s) to go to their closest fire*", and so is considered a soft-coded intervention, as the agent and fire locations remain dynamic. Figure 5 shows an abbreviation of the prompt template.

## 4.2 NATURAL LANGUAGE CONTROLLER

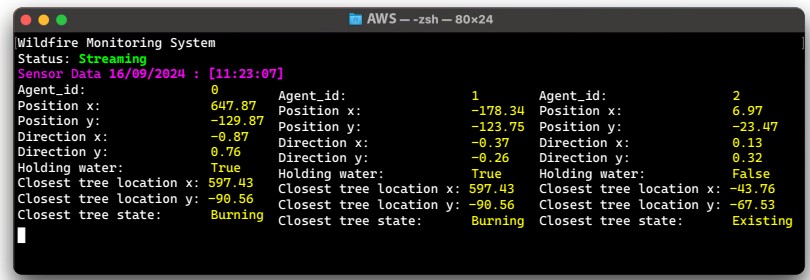

Figure 6: Possible AWS terminal as part of a fire-fighter dashboard. Info in this terminal is partially included in the NL strategy prompt template.

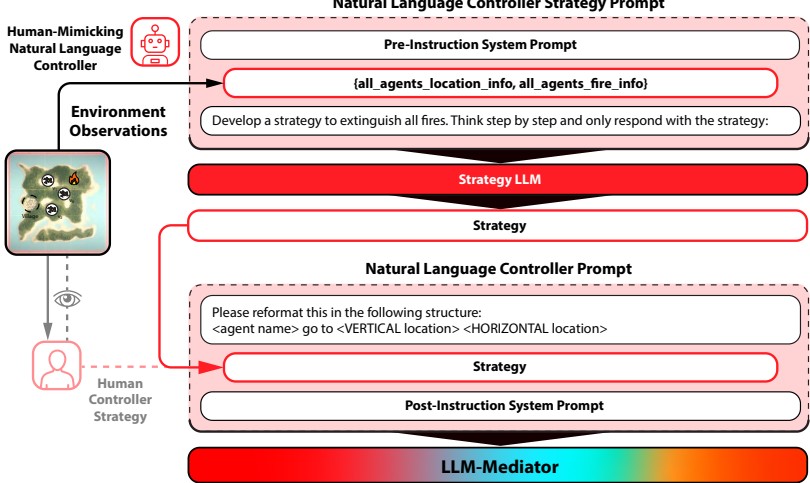

Figure 7: Abbreviated Natural Language Controller intervention prompts: 1. Human and Human-Mimicking LLM strategy prompt template generating strategies 2. A strategy as part of the prompt template is sent to the LLM-Mediator. A complete version can be found in the Appendix 13.

The NL Controller uses a prompt template with partial feature vector observation data (Figure 6). This information is provided as a list of all agents' observations and descriptions in natural language (Figure 7). The observation information formatted prompt is provided to an LLM, mimicking human behaviour, which generates a strategy directing agents to specific map locations. The NL Controller's high level directive is to "*Develop a strategy to extinguish all fires*". The resulting strategy is then passed to the LLM-Mediator. Matching with the Rule-Based Controller, the LLM-Mediator processes this more sophisticated strategy and returns agent-readable actions.

## 4.3 MEDIATOR

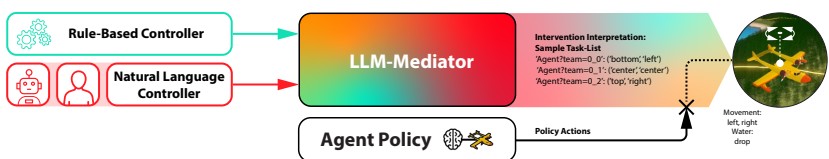

Figure 8: Rule-Based or Natural Language Controller interventions sent to LLM-Mediator, overwriting the agents' policy actions.

At the core, controllers act as prompt crafters. When a controller intervention prompt is issued, it is sent to the LLM-Mediator. Once the LLM-Interpreter processes the intervention, a task list is generated for each agent, and a 300-time-step cooldown period begins. During this period, agents are assigned their first task, and actions are generated to guide them toward task completion. These actions overwrite the agents' policy actions, such as steering left or right. If the agent holds water during the intervention period, the LLM-Mediator ensures it is retained by default. Each task includes a key to identify the agent and specify a target location (Figure 8). As long as the target location is not reached, actions continue to be auto-generated and issued to the agent. If the task is not completed within 300 time steps, a new intervention can be triggered. Figure 6 illustrates a basic terminal interface, as we imagine a human controller or firefighter using it to review observations, in combination with camera feed and radar data, etc., to determine whether an intervention should be issued.

## 4.4 PSEUDOCODE: MARL WITH LLM INTERVENTIONS

Our algorithm leverages a shared policy ($\pi_\theta$) for all agents, enabling simultaneous learning through centralized training. Experiences from all agents update the shared parameters. The LLM-Mediator selectively overrides agent actions based on cooldowns, while all collected experiences contribute to a single policy update, ensuring coordinated learning across agents. More details can be found in the code provided as well as the pseudocode in Algorithm 1.

## 5 EXPERIMENTS

To evaluate the effectiveness of RB and NL Controller interventions in our MARL framework, we conducted experiments within a custom AWS environment, part of the HIVEX suite. The experiments were designed to compare agents' performance under three different intervention setups: No Controller, RB and NL Controller. For LLMs, we used Pharia-1-LLM-7B-control-aligned (AlephAlpha, 2024) or Llama-3.1-8B Instruct (Meta, 2023). Experiments assess how well intervention and non-intervention-supported agents can learn and perform. All experiment setups utilize Proximal Policy Optimization (PPO) as the MARL algorithm (Schulman et al., 2017) and are trained on $3 \cdot 10^5$ time-steps. We use the default task (0) and terrain elevation level (1) of the AWS environment, but re-shaped rewards to focus on maximizing extinguishing tree rewards. We re-shaped the pick-up water reward from 1 to 0.1, the max preparing trees reward from 1 to 0.1 per tree, fire out reward from 10 to 0, too close to village reward from $-50$ to 0, and the max extinguishing trees reward from 5 to 1000 per tree.

---

**Algorithm 1** Multi-Agent RL with LLM Interventions and Cooldown Timers

---

Input: Multi-agent environment, PPO policy $\pi_\theta$, LLM-Mediator, intervention frequency $f$
Initialize environment, cooldown timers $\{c^i \leftarrow f\}_{i=1}^N$ for all agents $i$ and policy parameters $\theta_0$
**for** $episode = 1, 2, \ldots$ **do**
    Reset environment and cooldown timers $\{c^i \leftarrow f\}_{i=1}^N$
    **while** not done **do**
        Collect observations $\{s_t^i\}_{i=1}^N$ for all agents $i \in \{1, \ldots, N\}$
        Compute actions $\{a_t^i\}_{i=1}^N$ using policy $\pi_\theta(s_t^i)$ for each agent $i$
        **for** each agent $i \in \{1, \ldots, N\}$ **do**
            **if** $c^i == f$ **then**
                Generate intervention using LLM-Mediator:
                    $a_t^i \leftarrow$ LLM-Mediator$(s_t^i)$
                Reset cooldown timer for agent $i$: $c^i \leftarrow f$
            **else if** agent is currently following an LLM task **then**
                Decrement cooldown timer: $c^i \leftarrow c^i - 1$
                **if** $c^i < 0$ **then**
                    Reset cooldown timer: $c^i \leftarrow f$
                **end if**
            **end if**
        **end for**
        Perform a single step in the environment:
            $\{s_{t+1}^i, r_t^i\}_{i=1}^N \leftarrow$ env.step$(\{a_t^i\}_{i=1}^N)$
        Store transitions $\{(s_t^i, a_t^i, r_t^i, s_{t+1}^i)\}_{i=1}^N$ for all agents
    **end while**
    Update PPO policy $\pi_\theta$:
        Combine transitions from all agents into a shared buffer
        Compute advantage estimates $\{\hat{A}_t^i\}_{i=1}^N$ and rewards-to-go $\{\hat{R}_t^i\}_{i=1}^N$
        Optimize PPO objective to get $\theta_{k+1}$
**end for**

---

## 6 RESULTS

Our results show that the RB and NL Controller interventions outperform the baseline without interventions, highlighting the potential of LLM-mediated guidance to accelerate training and enhance MARL performance in challenging environments. Generally, we can say that intervention is better than none, even with sparse supervision. In addition, both intervention controllers achieve a high-performance level and adapt to the demands of the new environment directive. Table 1 shows performance on extinguishing trees reward and episode mean reward for three controller setups: None, RB and NL for Pharia-1-7B-control-aligned and LLama-3.1-8B Instruct. In Figure 10, we show mean *Extinguishing Trees Reward* and in Figure 9 *Episode Reward Mean* over 10 trials for each controller experiment, RB and NL versus the baseline without interventions. Please see Appendix A.7 for additional results.

We also investigated the scalability of our method by extending the default three-agent setup to

Table 1: No controller, RB and NL Controller performance on *Episode Reward Mean*[1] and *Extinguishing Trees Reward*[2] for Llama-3.1-8B Instruct and Pharia-1-LLM-control-aligned. *Average Wall-Time* per training run is in hour(s)[3].

| Mediator | Size | Controller | Episode R. Mean[1] | Ext. Trees R.[2] | Wall-Time[3] |
|---|---|---|---|---|---|
| | | None | 238.34 ($\pm$14.34) | 1.18 ($\pm$0.16) | **2.65** |
| Pharia-1-LLM | 7B | Rule-Based | **437.65** ($\pm$43.28) | 13.75 ($\pm$1.38) | 2.96 |
| | | Natural Language | 372.05 ($\pm$24.45) | 5.89 ($\pm$0.79) | 3.988 |
| Llama-3.1 | 8B | Rule-Based | 376.18 ($\pm$21.98) | **15.76** ($\pm$1.76) | 3.13 |
| | | Natural Language | 331.22 ($\pm$39.88) | 6.73 ($\pm$0.81) | 5.72 |

configurations with four, five, and six agents. Performance was compared between RB interventions and the no-intervention baseline using *Episode Reward Mean* and *Extinguishing Trees Reward Mean* for Pharia-1-7B-control-aligned and LLama-3.1-8B Instruct (Figure 11).

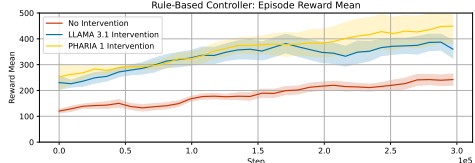 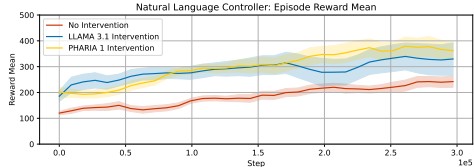

Figure 9: Episode Reward Mean: Left: No controller baseline VS Rule-Based Controller with Llama-3.1-8B Instruct and Pharia-1-LLM-control-aligned-Mediator. Right: No controller baseline VS Natural Language Controller with Llama-3.1-8B Instruct and Pharia-1-LLM-7B-control-aligned-Mediator.

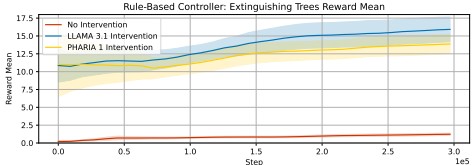 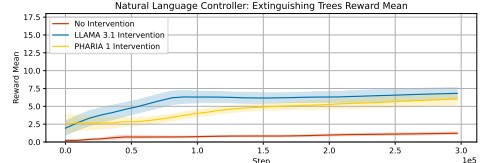

Figure 10: Extinguishing Trees Reward Mean: Left: No controller baseline VS Rule-Based Controller with Llama-3.1-8B Instruct and Pharia-1-LLM-control-aligned-Mediator. Right: No controller baseline VS Natural Language Controller with Llama-3.1-8B Instruct and Pharia-1-LLM-control-aligned-Mediator.

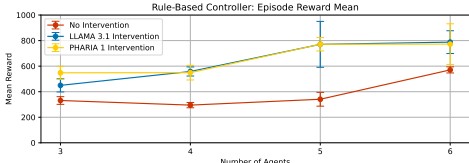 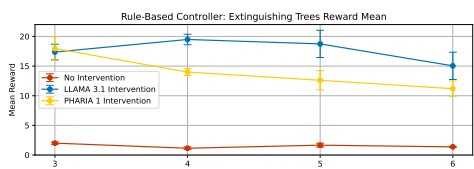

Figure 11: Scalability Experiment with 3 (default), 4, 5 and 6 agents: No controller baseline VS Rule-Based Controller with Llama-3.1-8B Instruct and Pharia-1-LLM-control-aligned-Mediator: Episode Reward Mean (left), Extinguishing Trees Reward Mean (right).

## 7 DISCUSSION

The results of our experiments provide valuable insights into the effectiveness of LLM-based interventions in MARL. Our findings show that periodic interventions, mimicking human behaviour, can significantly enhance agents' performance in complex environments like AWS, where coordinated actions across multiple agents are crucial.

A key observation is the comparative advantage of NL Controller interventions over non-intervention baselines. Pharia-1-LLM-7B-control-aligned outperformed in the Rule-Based Environment Mean Rewards, while Llama-3.1-8B Instruct excelled in the Extinguishing Trees Reward category. This suggests that Pharia-1-LLM-7B-control-aligned handles structured interventions better, while Llama-3.1-8B Instruct is more adept at free-form natural language interventions. The 300-step intervention cooldown allowed agents to consolidate learning, operating independently for approximately 10 steps. The adaptability of LLMs in real-time, context-sensitive guidance is evident, though each model excels in different dimensions. Both would benefit from memory of past tasks to refine strategies and enhance their adaptability in rapidly changing environments. The scalability experiments show that RB interventions consistently outperform the no-intervention baseline as agent numbers increase. Pharia-1 slightly outperforms LLama-3.1 in Episode Reward Mean,

while both show a small decline in Extinguishing Trees Reward Mean with more agents, indicating coordination challenges in larger teams.

These findings suggest that LLM-based NL Controller interventions offer a promising approach for improving MARL systems, particularly where traditional RL methods face limitations. The distinct strengths of Pharia-1-LLM-7B-control-aligned and Llama-3.1-8B Instruct underscore the need for continued research to enhance LLM reasoning and planning capabilities. Further studies in more realistic environments are needed to validate these results across different domains.

## 8 LIMITATIONS AND POTENTIAL IMPACTS

While our research demonstrates the significant potential of integrating LLMs into MA systems, several limitations and considerations must be acknowledged, particularly concerning bias, safety, the realism of the environment, and the transferability of our findings to other domains. Further discussion and information on resources and inference cost, and bias and safety concerns can be found in the Appendix A.1.

**Realism of the Environment**: One limitation is the realism of the experimental environment. Although the AWS environment simulates real-world challenges, discrepancies remain between the simulation, actual wildfire scenarios, and the control mechanisms of autonomous aeroplanes. These differences may affect the generalizability of our findings, as agents trained in a simulated setting may underperform in real-world conditions. Moreover, fine-tuning the models using real-world data could be costly. Enhancing the simulation to mirror real-world conditions and incorporating additional realistic variables more closely would help mitigate this limitation.

**Transferability to Other Domains**: Our LLM-Mediator approach's success in the AWS environment context raises questions about its transferability to other domains. While the adaptive and context-sensitive nature of LLM-mimicked human interventions shows promise, different tasks and environments may require tailored adjustments to achieve similar levels of effectiveness. The complexity of the task, the nature of agent interactions, and the specific challenges of the domain in question all influence how well this approach can be applied elsewhere. Future research should explore the adaptability of intervention and LLM-driven mediation across various MARL applications to investigate its broader applicability.

**Potential Impacts**: Despite these limitations, the potential impacts of our research are substantial. By demonstrating the effectiveness of intervention and LLM-driven mediation in accelerating learning and improving coordination among agents, our approach offers a scalable solution for enhancing MARL systems in complex, dynamic environments. The findings suggest that human-like reasoning can lead to more efficient and effective learning processes, potentially reducing the computational resources required to train agents in complex environments. As these methods are refined and adapted to other domains, they could significantly advance the field of RL, contributing to more resilient and intelligent MA systems capable of tackling a wide range of real-world challenges.

## 9 CONCLUSION

This paper demonstrates the potential of integrating LLMs into MARL environments, particularly in interpreting complex environmental observations and mediating real-time, context-sensitive interventions. Our experiments within the MA Aerial Wildfire Suppression environment part of the HIVEX suite show that periodic LLM guidance significantly improves agent performance, surpassing rule-based and non-guided baselines. Pharia-1-LLM-7B-control-aligned excelled in structured, rule-based tasks, while Llama-3.1-8B Instruct performed better in dynamic, situational challenges, highlighting the complementary strengths of different LLMs as mediators. This work underscores the scalability and efficiency of LLMs, particularly when mimicking human expertise, as a promising alternative to direct human guidance.

In conclusion, our findings suggest that LLMs and MARL techniques have matured to a point where they can effectively adapt systems to complex, dynamic environments —an essential capability for tackling real-world challenges. The versatility of LLM-mediated interventions allows for easy adaptation to other domains, enabling efficient 'fine-tuning' of MARL systems for specific tasks. While fully automating curriculum design remains challenging, minimal real-time human supervision can provide cost-effective, sparse guidance, helping agents develop more efficient policies and address increasingly complex tasks.

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

# A APPENDIX

## A.1 ADDITIONAL LIMITATIONS AND POTENTIAL IMPACTS

**Bias and Safety Concerns**: A key limitation of using LLMs is the risk of bias in their human-mimicked interventions, stemming from the potentially biased datasets they are trained on. Such biases could result in suboptimal or harmful behaviours, particularly in critical tasks like wildfire suppression. Additionally, deploying LLMs in real-world environments raises safety concerns due to unpredictable outcomes. Rigorous testing and validation in controlled settings are essential to mitigate these risks.

**Resources and Inference Cost**: Another important consideration is the inference cost associated with the human LLM-mimicked interventions and LLM-Mediator. Out of the 3000 total steps per agent per episode, the inference cost is only a fraction, as interventions are introduced every 300 steps and typically influence agent behaviour for approximately $\sim 200$ steps. This periodic intervention minimizes the computational overhead, allowing agents to continue operating efficiently under the learned policy for the remaining 100 steps. By balancing intervention frequency and task completion duration, we ensure that the computational load is manageable while still leveraging the benefits of real-time guidance from LLMs. Future work could further explore optimising this balance, reducing the task completion duration or intervention frequency while maintaining or improving agent performance. The training and testing of our experiment have been conducted on accessible, end-user hardware featuring an NVIDIA GeForce RTX 3090 GPU, an AMD Ryzen 9 7950X 16-Core Processor, and 64 GB of RAM. While these specifications align with high-end gaming laptops and desktop computers, the configuration could still be adapted to low-budget and non-GPU environments. This eliminates the need for specialized computational clusters, ensuring that researchers and practitioners with mid-range to high-end hardware can readily replicate our results using only consumer-grade equipment and an API for the LLM-Mediator.

## A.2 PSEUDOCODE

Standard PPO-CLIP pseudocode (OpenAI, 2021; Schulman et al., 2017):

---

**Algorithm 2**

---

Input: initial policy parameters $\theta_0$, initial value function parameters $\phi_0$

**for** $k = 0, 1, 2, \ldots$ **do**

    Collect set of trajectories $\mathcal{D}_k = \{\tau_i\}$ by running policy $\pi_k = \pi(\theta_k)$ in the environment and overwriting with LLM-Mediator generated actions if an intervention has been issued.

    Compute rewards-to-go $\hat{R}_t$.

    Compute advantage estimates, $\hat{A}_t$ (using any method of advantage estimation) based on the current value function $V_{\phi_k}$

    Update the policy by maximizing the PPO-Clip objective:

$$\theta_{k+1} = arg\max_{\theta} \frac{1}{|\mathcal{D}_k|T} \sum_{\tau \in \mathcal{D}_k} \sum_{t=0}^{T} \min\left( \frac{\pi_\theta(a_t|s_t)}{\pi_{\theta_k}(a_t|s_t)} A^{\pi_{\theta_k}}(s_t, a_t), g(\epsilon, A^{\pi_{\theta_k}}(s_t, a_t)) \right),$$

    typically via stochastic gradient ascent with Adam.

    Fit value function by regression on mean-squared error:

$$\phi_{k+1} = arg\min_{\phi} \frac{1}{|\mathcal{D}_k|T} \sum_{\tau \in \mathcal{D}_k} \sum_{t=0}^{T} \left( (V_\phi(s_t) - \hat{R}_t \right)$$

    typically via some gradient descent algorithm.

**end for**

---

## A.3 HYPERPARAMETERS

### A.3.1 NO INTERVENTION

```
name: "NO_INTERVENTION"
env_parameters:
  training: 1
  human_intervention: 0
  task: 0
  ext_fire_reward: 1000
  prep_tree_reward: 0.1
  water_pickup_reward: 0.1
  fire_out_reward: 0
  crash_reward: -100
  fire_close_to_city_reward: 0
no_graphics: True
intervention_type: "none"
lr: 0.005
lambda_: 0.95
gamma: 0.99
sgd_minibatch_size: 900
train_batch_size: 9000
num_sgd_iter: 3
clip_param: 0.2
```

### A.3.2 RULE-BASED LLAMA-3.1-8B INSTRUCT

```
name: "RB_LLAMA_3.1"
env_parameters:
  training: 1
  human_intervention: 0
  task: 0
  ext_fire_reward: 1000
  prep_tree_reward: 0.1
  water_pickup_reward: 0.1
  fire_out_reward: 0
  crash_reward: -100
  fire_close_to_city_reward: 0
no_graphics: True
intervention_type: "auto"
model: "llama-3.1-8b-instruct"
shot: "few"
lr: 0.005
lambda_: 0.95
gamma: 0.99
sgd_minibatch_size: 900
train_batch_size: 9000
num_sgd_iter: 3
clip_param: 0.2
```

### A.3.3 RULE-BASED PHARIA-1-LLM-7B-CONTROL-ALIGNED

```
name: "RB_PHARIA_1"
env_parameters:
  training: 1
  human_intervention: 0
  task: 0
  ext_fire_reward: 1000
  prep_tree_reward: 0.1
  water_pickup_reward: 0.1
  fire_out_reward: 0
  crash_reward: -100
  fire_close_to_city_reward: 0
no_graphics: True
intervention_type: "auto"
model: "Pharia-1-LLM-7B-control-aligned"
shot: "few"
lr: 0.005
lambda_: 0.95
gamma: 0.99
sgd_minibatch_size: 900
train_batch_size: 9000
num_sgd_iter: 3
clip_param: 0.2
```

### A.3.4 NATURAL LANGUAGE LLAMA-3.1-8B INSTRUCT

```
name: "NL_LLAMA_3.1"
env_parameters:
  training: 1
  human_intervention: 0
  task: 0
  ext_fire_reward: 1000
  prep_tree_reward: 0.1
  water_pickup_reward: 0.1
  fire_out_reward: 0
  crash_reward: -100
  fire_close_to_city_reward: 0
no_graphics: True
intervention_type: "llm"
model: "llama-3.1-8b-instruct"
shot: few
lr: 0.005
lambda_: 0.95
gamma: 0.99
sgd_minibatch_size: 900
train_batch_size: 9000
num_sgd_iter: 3
clip_param: 0.2
```

### A.3.5 NATURAL LANGUAGE PHARIA-1-LLM-7B-CONTROL-ALIGNED

```
name: "NL_PHARIA_1"
env_parameters:
  training: 1
  human_intervention: 0
  task: 0
  ext_fire_reward: 1000
  prep_tree_reward: 0.1
  water_pickup_reward: 0.1
  fire_out_reward: 0
  crash_reward: -100
  fire_close_to_city_reward: 0
no_graphics: True
intervention_type: "llm"
model: "Pharia-1-LLM-7B-control-aligned"
shot: few
lr: 0.005
lambda_: 0.95
gamma: 0.99
sgd_minibatch_size: 900
train_batch_size: 9000
num_sgd_iter: 3
clip_param: 0.2
```

## A.4 ENVIRONMENT SPECIFICATION

- Episode Length: 3000
- Agent Count: 3
- Neighbour Count: 0

**Feature Vector Observations (8)** - Stacks: 1 - Normalized: True

- Local Position (2): $\vec{p}(x, y)$
- Direction (2): $\vec{dir}(x, y)$
- Holding Water (1): $hw = [0, 1]$
- Closest Tree Location (2): $\vec{ct}(x, y)$
- Closest Tree Burning (1): $ctb = [0, 1]$

**Visual Observations (42, 42, 3)** - Stacks: 1 - Normalized: True

- Downward Pointing Camera in RGB (1764): $[r, g, b] = [[0, 1], [0, 1], [0, 1]]$

**Continous Actions (1)**:

- Steer Left / Right (1): $[-1, 1]$

**Discrete Actions (1)**:

- Branch 0 - Drop Water (2): 0: Do Nothing, 1: Drop Water

## A.5 Un-shaped Reward Description and Calculation

**Reward Description**

1. **Crossed Border** - This is a negative reward of $-100$ given when the border of the environment is crossed. The border is a square around the island in the size of $1500$ by $1500$. The island is $1200$ by $1200$.
2. **Pick-up Water** - This is a positive reward of $1$ given when the agent steers the aeroplane towards the water. The island is $1200$ by $1200$ and there is a girdle of water around the island with a width of $300$.
3. **Fire Out** - This is a positive reward of $10$ given when the fire on the whole island dies out, with or without the active assistance of the agent.
4. **Too Close to Village** - This is a negative reward of $-50$ given when the fire is closer than $150$ to the centre of the village.
5. **Time Step Burning** - This is a negative reward of $-0.01$ given at each time-step, while the fire is burning.
6. **Extinguishing Tree** - This is a positive reward in the range of $[0, 5]$ given for each tree that has been in the state burning in time-step $t_{-1}$ and is now extinguished by dropping water at its location.
7. **Preparing Tree** - This is a positive reward in the range of $[0, 1]$ given for each tree that has been in the state not burning in time-step $t_{-1}$ and is now wet by dropping water at its location.

**Reward Calculation**

**1. Crossed Border** - To calculate the Crossed Border reward, let us define the following:

- $eh = 750$ — The environment half extend.
- $\vec{p}$ — The drone position.
- $r_{cb}$ — Crossed boundary reward.

Calculation steps:

1. We can now calculate the Crossed Border reward:

$$r_{cb} = \begin{cases} -100 & \text{if } (p_x > eh \text{ or } p_x < -eh \text{ or } p_y > eh \text{ or } p_y < -eh) \\ 0 & \text{otherwise} \end{cases} \quad (1)$$

**2. Pick-up Water** - To calculate the Pick-up Water reward, let us define the following:

- $eh = 750$ — The environment half extend.
- $ih = 600$ — Island half extend.
- $\vec{p}$ — The drone position.
- $r_{pw}$ — Pick-up Water reward.

Calculation steps:

1. We can now calculate the Pick-up Water reward:

$$r_{pw} = \begin{cases} 1 & \text{if } (p_x < eh \text{ or } p_x > -eh \text{ or } p_y < eh \text{ or } p_y > -eh) \\ & \text{and } (p_x > ih \text{ or } p_x < -ih \text{ or } p_y > ih \text{ or } p_y < -ih) \\ 0 & \text{otherwise} \end{cases} \quad (2)$$

**3. Fire Out** - To calculate the Fire Out reward, let us define the following:

- $T$ — All tree states.
- $r_{nb}$ — No burning tree reward.

Calculation steps:

1. We can now calculate the Fire Out reward:

$$r_{nb} = \begin{cases} 10 & \text{if } \forall t \in T, \ t \neq \text{"burning"} \\ 0 & \text{otherwise} \end{cases} \quad (3)$$

**4. Too Close to Village** - To calculate the Too Close to Village reward, let us define the following:

- $T_c$ — All tree states, closer to or equal to $150$ to the village.
- $r_{cv}$ — Too Close to Village reward.

Calculation steps:

1. We can now calculate the Fire Out reward:

$$r_{cc} = \begin{cases} -50 & \text{if } \exists t \in T_c,\ t = \text{"burning"} \\ 0 & \text{otherwise} \end{cases} \tag{4}$$

**5. Time Step Burning** - To calculate the Time Step Burning reward, let us define the following:

- $T$ — All tree states.
- $r_{tsb}$ — Time Step Burning reward.

Calculation steps:

1. We can now calculate the Time Step Burning reward:

$$r_{tsb} = \begin{cases} -0.01 & \text{if } \forall t \in T,\ t = \text{"burning"} \\ 0 & \text{otherwise} \end{cases} \tag{5}$$

**6. Extinguishing Tree** - To calculate the Extinguish Tree reward, let us define the following:

- $T$ — All tree states.
- $r_e$ — Extinguish Tree reward.

Calculation steps:

1. We can now calculate the Extinguish Tree reward:

$$r_e = 5 \sum_{t \in T} \mathbb{I}(t_{\text{previous}} = \text{"burning" and } t_{\text{current}} = \text{"extinguished"}) \tag{6}$$

**7. Preparing Tree** - To calculate the Preparing Tree reward, let us define the following:

- $T$ — All tree states.
- $r_p$ — Preparing Tree reward.

Calculation steps:

1. We can now calculate the Preparing Tree reward:

$$r_e = \sum_{t \in T} \mathbb{I}(t_{\text{previous}} = \text{"not Burning" and } t_{\text{current}} = \text{"wet"}) \tag{7}$$

## A.6 PROMPT TEMPLATES & SAMPLES

### A.6.1 RULE-BASED CONTROLLER PROMPT TEMPLATE: LLM-MEDIATOR

**Rule-Based Controller Prompt**

**Pre-Instruction System Prompt**

<|begin_of_text|><|start_header_id|>user<|end_header_id|>
Instruction:
HORIZONTAL locations: ['left', 'right', 'center']
VERTICAL locations: ['bottom', 'top', 'center']
Instruct agent(s) to go to their closest fire.
Reply like this: <agent name> go to <VERTICAL location> <HORIZONTAL location>
Input:
Agent 'Agent?team=0_0' is in the bottom center, Agent 'Agent?team=0_1' is in the center right,
Agent 'Agent?team=0_2' is in the bottom center. 1 fire(s): Fire 0 is in the center center.
Where should agents go?<|eot_id|><|start_header_id|>assistant<|end_header_id|>
Agent 'Agent?team=0_0' go to center center
Agent 'Agent?team=0_1' go to center center
Agent 'Agent?team=0_2' go to center center<|eot_id|><|start_header_id|>user<|end_header_id|>
Instruction:
HORIZONTAL locations: ['left', 'right', 'center']
VERTICAL locations: ['bottom', 'top', 'center']
Instruct agent(s) to go to their closest fire.
Reply like this: <agent name> go to <VERTICAL location> <HORIZONTAL location>
Input:

**Rule-Based Intervention: {all_agents_location_info, all_agents_fire_info}**

**Post-Instruction System Prompt**

Where should agents go?<|eot_id|><|start_header_id|>assistant<|end_header_id|>

**Environment Observations**

**LLM-Mediator**

Figure 12: Complete prompt template for the Rule-Based Controller. This prompt is sent to the LLM-Mediator.

## A.6.2 NATURAL LANGUAGE CONTROLLER PROMPT TEMPLATE: STRATEGY AND LLM-MEDIATOR

Figure 13: Complete prompt templates for the Natural Language Controller. The first prompt template is to generate a strategy, which is then integrated in the second prompt template that is sent to the LLM-Mediator.

### A.6.3 RULE-BASED AND NATURAL LANGUAGE CONTROLLER VECTOR OBSERVATION DATA SAMPLES

**{all_agents_location_info}**
**{all_agents_fire_info}**
Samples for Rule-Based and Natural Language Controller Prompt Templates

Sample 1
"all_agents_location_info":
**"Agent 'Agent?team=0_0' is in the center right, Agent 'Agent?team=0_1' is in the top right, Agent 'Agent?team=0_2' is in the top center."**
"all_agents_fire_info":
**"1 fire(s): Fire 0 is in the top right."**

Sample 2
"all_agents_location_info":
**"Agent 'Agent?team=0_0' is in the center center, Agent 'Agent?team=0_1' is in the bottom left, Agent 'Agent?team=0_2' is in the center center."**
"all_agents_fire_info":
**"1 fire(s): Fire 0 is in the center right."**

Sample 3
"all_agents_location_info":
**"Agent 'Agent?team=0_0' is in the top left, Agent 'Agent?team=0_1' is in the center right, Agent 'Agent?team=0_2' is in the center center."**
"all_agents_fire_info":
**"1 fire(s): Fire 0 is in the center right."**

Sample 4
"all_agents_location_info":
**"Agent 'Agent?team=0_0' is in the top center, Agent 'Agent?team=0_1' is in the center center, Agent 'Agent?team=0_2' is in the center left."**
"all_agents_fire_info":
**"1 fire(s): Fire 0 is in the top center."**

Sample 5
"all_agents_location_info":
**"Agent 'Agent?team=0_0' is in the bottom center, Agent 'Agent?team=0_1' is in the top left, Agent 'Agent?team=0_2' is in the top right."**
"all_agents_fire_info":
**"1 fire(s): Fire 0 is in the top center."**

Sample 6
"all_agents_location_info":
**"Agent 'Agent?team=0_0' is in the center center, Agent 'Agent?team=0_1' is in the bottom center, Agent 'Agent?team=0_2' is in the center left."**
"all_agents_fire_info":
**"1 fire(s): Fire 0 is in the center center."**

Sample 7
"all_agents_location_info":
**"Agent 'Agent?team=0_0' is in the bottom left, Agent 'Agent?team=0_1' is in the top center, Agent 'Agent?team=0_2' is in the top center."**
"all_agents_fire_info":
**"1 fire(s): Fire 0 is in the top right."**

Sample 8
"all_agents_location_info":
**"Agent 'Agent?team=0_0' is in the top left, Agent 'Agent?team=0_1' is in the top right, Agent 'Agent?team=0_2' is in the center center."**
"all_agents_fire_info":
**"1 fire(s): Fire 0 is in the top right."**

Sample 9
"all_agents_location_info":
**"Agent 'Agent?team=0_0' is in the top right, Agent 'Agent?team=0_1' is in the top right, Agent 'Agent?team=0_2' is in the top right."**
"all_agents_fire_info":
**"1 fire(s): Fire 0 is in the top right."**

Sample 10
"all_agents_location_info":
**"Agent 'Agent?team=0_0' is in the center center, Agent 'Agent?team=0_1' is in the bottom right, Agent 'Agent?team=0_2' is in the top center."**
"all_agents_fire_info":
**"1 fire(s): Fire 0 is in the top center."**

Figure 14: Feature Vector observation data samples in natural language of {all_agents_location_info} and {all_agents_fire_info}, integrated in the Rule-Based Controller prompt template as well as the strategy prompt template as part of the Natural Language Controller.

### A.6.4 NATURAL LANGUAGE STRATEGY SAMPLES: PHARIA-1-LLM-7B-CONTROL-ALIGNED

{strategy}
Samples for Natural Language Controller Strategy Prompt Completion using **Pharia-1-LLM 7B Control Aligned**

**Sample 1**
HORIZONTAL VERTICAL AGENT 0_0 CENTER LEFT
HORIZONTAL AGENT 0_0 CENTER CENTER
HORIZONTAL AGENT 0_0 CENTER RIGHT

VERTICAL AGENT 0_0 BOTTOM LEFT
VERTICAL AGENT 0_0 TOP RIGHT
VERTICAL AGENT 0_0 TOP CENTER

HORIZONTAL AGENT 0_1 BOTTOM LEFT
HORIZONTAL AGENT 0_1 CENTER LEFT
HORIZONTAL AGENT 0_1 CENTER RIGHT

VERTICAL AGENT 0_1 BOTTOM CENTER
VERTICAL AGENT 0_1 CENTER RIGHT
VERTICAL AGENT 0_1 TOP RIGHT

HORIZONTAL AGENT 0_2 TOP CENTER
HORIZONTAL AGENT 0

**Sample 2**
 HORIZONTAL VERTICAL AGENT 0_0 LEFT AGENT 0_1 RIGHT AGENT 0_2 RIGHT

* HORIZONTAL AGENT 0_0 LEFT: Move to the top left corner.
* HORIZONTAL AGENT 0_1 RIGHT: Move to the top right corner.
* HORIZONTAL AGENT 0_2 RIGHT: Move to the bottom right corner.

* VERTICAL AGENT 0_0 LEFT: Move to the top center.
* VERTICAL AGENT 0_1 RIGHT: Move to the top center.
* VERTICAL AGENT 0_2 RIGHT: Move to the bottom center.

This strategy will ensure that all agents are in the vicinity of the fire and can quickly extinguish or prepare the trees for incoming fire. The agents will

**Sample 3**
 HORIZONTAL VERTICAL AGENT 'Agent?team=0_1'
  1. Bottom left
  2. Bottom center
  3. Bottom right

HORIZONTAL AGENT 'Agent?team=0_2'
  1. Top left
  2. Top center
  3. Top right

VERTICAL AGENT 'Agent?team=0_0'
  1. Top center
  2. Top right
  3. Bottom right

The strategy is to have Agent 'Agent?team=0_1' extinguish Fire 0, Agent 'Agent?team=0_2' extinguish Fire 1, and Agent 'Agent?team=0_0' extinguish Fire 2.

HORIZONTAL AGENT 'Agent?team=0

**Sample 4**
 HORIZONTAL STRATEGY:

* Agent 'Agent?team=0_0': Move to the center right (Fire 0).
* Agent 'Agent?team=0_1': Move to the center right (Fire 0).

VERTICAL STRATEGY:

* Agent 'Agent?team=0_0': Move to the bottom right (Fire 0).
* Agent 'Agent?team=0_1': Move to the bottom right (Fire 0).

HORIZONTAL + VERTICAL STRATEGY:

* Agent 'Agent?team=0_0': Move to the bottom right (Fire 0).
* Agent 'Agent?team=0_1': Move to the bottom right (Fire 0).

* Agent 'Agent

Figure 15: Pharia-1-LLM-7B-control-aligned samples for {strategy}, to be integrated in the Natural Language Controller prompt template, sent to the LLM-Mediator.

### A.6.5 Natural Language Strategy Samples: Llama-3.1-8B Instruct

**{strategy}**
Samples for Natural
Language Controller Strategy
Prompt Completion using
**Llama-3.1-8B Instruct**

**Sample 1**
Based on the given positions and the fire location, a basic strategy for the agents could be:

1. Agent 'Agent?team=0_0' in the top center moves to the top right to surround the fire.
2. Agent 'Agent?team=0_1' in the center left moves to the top left to surround the fire.
3. Agent 'Agent?team=0_2' in the top left moves to the top center to extinguish the fire.

This strategy involves horizontal movements for Agent 'Agent?team=0_0' and Agent 'Agent?team=0_2', and vertical movements for Agent 'Agent?team=0_1'.

**Sample 2**
Based on the given positions of the agents and the fire location, a basic strategy to extinguish or prepare trees for the incoming fire could be as follows:

1. Agent 'Agent?team=0_0' is already in the center right position. It can move to the top right position (HORIZONTAL) to join Agent 'Agent?team=0_1' and prepare the area for the fire. This would be a more effective strategy than moving to the center center or center left positions.

2. Agent 'Agent?team=0_1' is already in the top right position. It can move to the top center position (HORIZONTAL) to join Agent 'Agent?team=0_2' and prepare the area for the fire.

3. Agent 'Agent?team=0_2' is already in the top center position. It can move to the top left position (HORIZONTAL) to prepare the area for the fire.

4. Once

**Sample 3**
Based on the given scenario, a basic strategy for the agents to extinguish or prepare trees for the incoming fire in the bottom right location could be as follows:

1. Agent 'Agent?team=0_0' moves to the center left position. This allows the agent to be in a central location and have the ability to move horizontally or vertically to the fire location.
2. Agent 'Agent?team=0_1' moves to the center right position. This places the agent in a position to move directly to the fire location or to the surrounding areas to prepare trees for the fire.
3. Agent 'Agent?team=0_2' moves to the bottom left position. This allows the agent to be in a position to move directly to the fire location or to the surrounding areas to prepare trees for the fire.

This strategy positions the agents in a way that they can quickly respond to the fire location and work together to extinguish or prepare trees

**Sample 4**
Based on the given positions of the agents and the fire location, a basic strategy to extinguish or prepare trees for the incoming fire could be:

1. Agent 'Agent?team=0_0' moves from the center center to the bottom center (HORIZONTAL) to directly engage the fire.
2. Agent 'Agent?team=0_1' moves from the center left to the bottom left (VERTICAL) to support Agent 'Agent?team=0_0' and prepare the surrounding area for the fire.
3. Agent 'Agent?team=0_2' moves from the center center to the bottom right (HORIZONTAL) to cover the right flank and prevent the fire from spreading.

This strategy involves a combination of horizontal and vertical movements to effectively address the fire location and prevent its spread.

**Sample 5**
Based on the given positions of the agents and the fire location, a basic strategy to extinguish or prepare trees for the incoming fire could be as follows:

1. Agent 'Agent?team=0_0' in the top center should move to the top right position to surround the fire from the right side. This is a horizontal movement from the center to the right.

2. Agent 'Agent?team=0_1' in the bottom center should move to the center right position to surround the fire from the right side. This is a vertical movement from the bottom to the center.

3. Agent 'Agent?team=0_2' in the center center should move to the top center position to surround the fire from the top side. This is a vertical movement from the center to the top.

By executing these movements, all agents will be positioned around the fire, effectively surrounding it and preparing the area for extinguishing or preparing trees for the incoming fire.

Figure 16: LLama-3.1-8B Instruct samples for {strategy}, to be integrated in the Natural Language Controller prompt template, sent to the LLM-Mediator.

## A.7 ADDITIONAL RESULTS

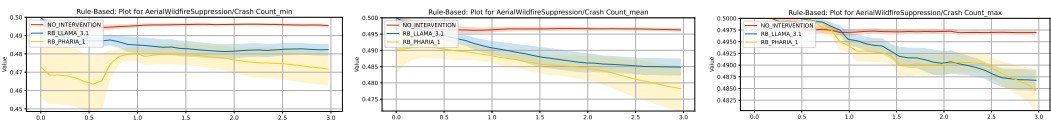

Figure 17: Crash Count (**Rule-Based**) - No controller baseline VS Rule-Based Controller with Llama-3.1-8B Instruct: min, mean and max.

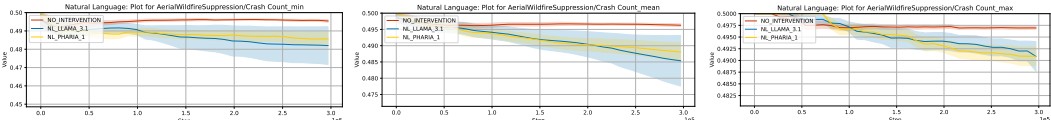

Figure 18: Crash Count (**Natural Language**) - No controller baseline VS Natural Language Controller with Llama-3.1-8B Instruct: min, mean and max.

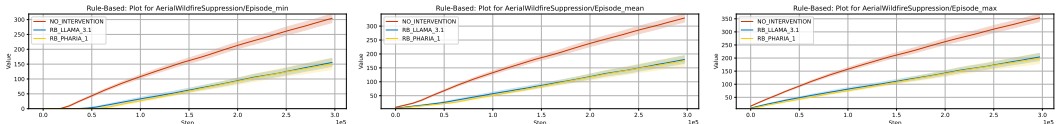

Figure 19: Episode Count (**Rule-Based**) - No controller baseline VS Rule-Based Controller with Llama-3.1-8B Instruct: min, mean and max.

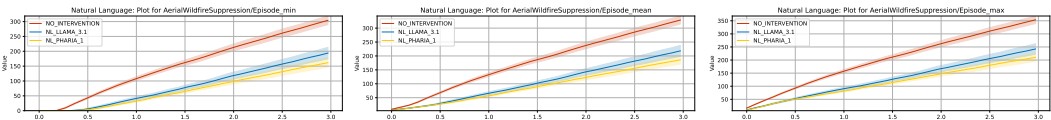

Figure 20: Episode Count (**Natural Language**) - No controller baseline VS Natural Language Controller with Llama-3.1-8B Instruct: min, mean and max.

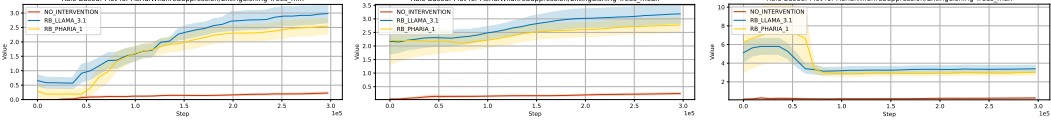

Figure 21: Extinguishing Trees (**Rule-Based**) - No controller baseline VS Rule-Based Controller with Llama-3.1-8B Instruct: min, mean and max.

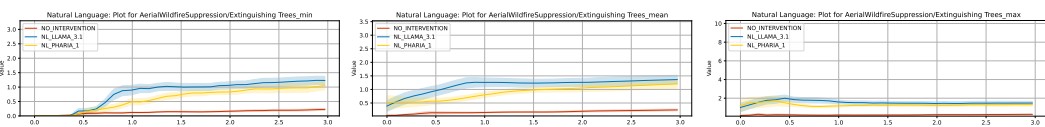

Figure 22: Extinguishing Trees (**Natural Language**) - No controller baseline VS Natural Language Controller with Llama-3.1-8B Instruct: min, mean and max.

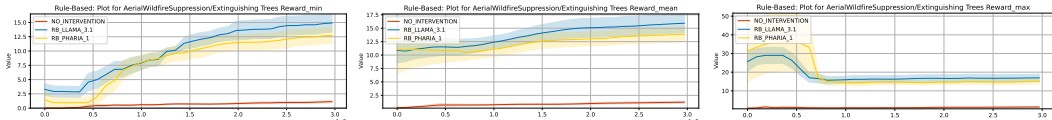

Figure 23: Extinguishing Trees Reward (**Rule-Based**) - No controller baseline VS Rule-Based Controller with Llama-3.1-8B Instruct: min, mean and max.

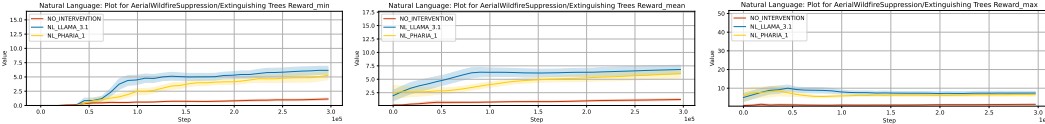

Figure 24: Extinguishing Trees Reward (**Natural Language**) - No controller baseline VS Natural Language Controller with Llama-3.1-8B Instruct: min, mean and max.

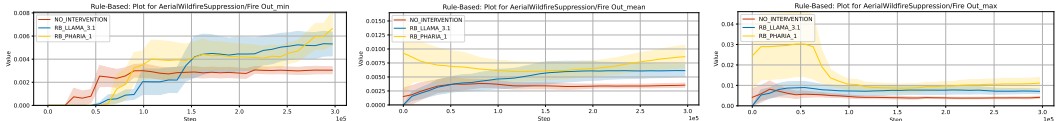

Figure 25: Fire Out Count (**Rule-Based**) - No controller baseline VS Rule-Based Controller with Llama-3.1-8B Instruct: min, mean and max.

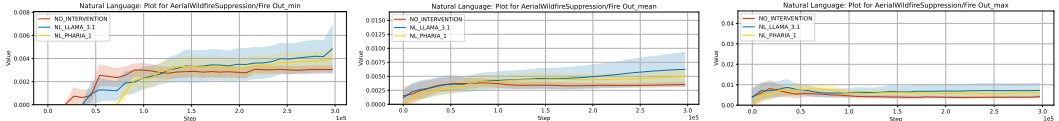

Figure 26: Fire Out Count (**Natural Language**) - No controller baseline VS Natural Language Controller with Llama-3.1-8B Instruct: min, mean and max.

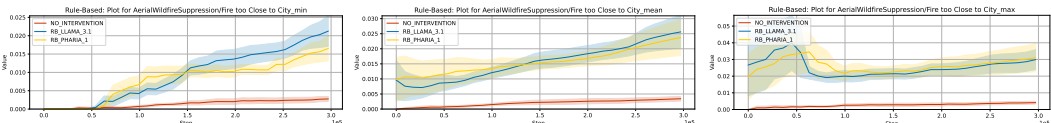

Figure 27: Fire too Close to City (**Rule-Based**) - No controller baseline VS Rule-Based Controller with Llama-3.1-8B Instruct: min, mean and max.

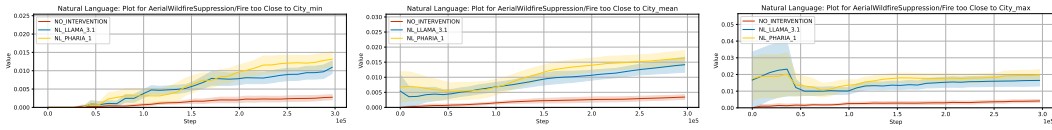

Figure 28: Fire too Close to City (**Natural Language**) - No controller baseline VS Natural Language Controller with Llama-3.1-8B Instruct: min, mean and max.

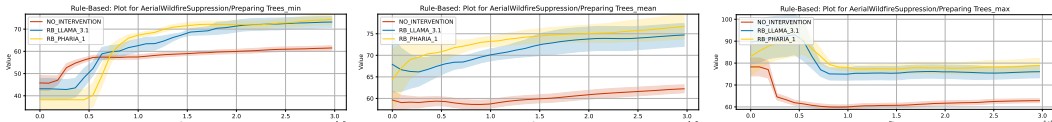

Figure 29: Preparing Trees (**Rule-Based**) - No controller baseline VS Rule-Based Controller with Llama-3.1-8B Instruct: min, mean and max.

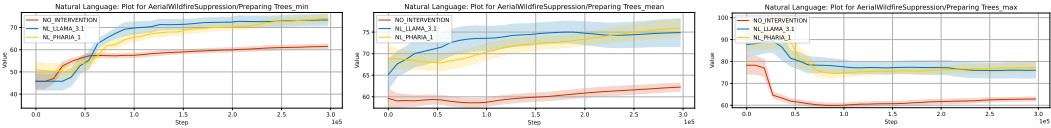

Figure 30: Preparing Trees (**Natural Language**) - No controller baseline VS Natural Language Controller with Llama-3.1-8B Instruct: min, mean and max.

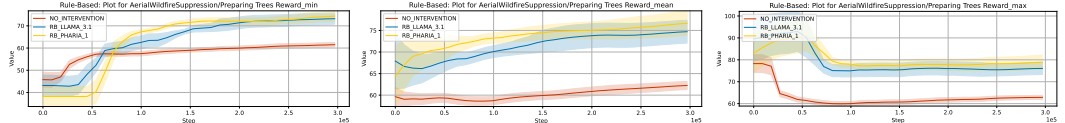

Figure 31: Preparing Trees Reward (**Rule-Based**) - No controller baseline VS Rule-Based Controller with Llama-3.1-8B Instruct: min, mean and max.

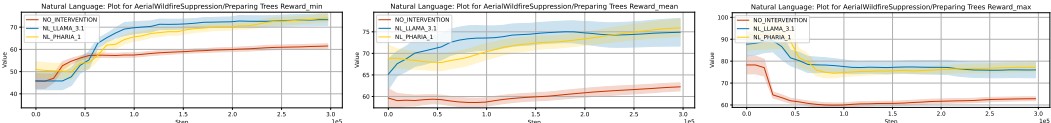

Figure 32: Preparing Trees Reward (**Natural Language**) - No controller baseline VS Natural Language Controller with Llama-3.1-8B Instruct: min, mean and max.

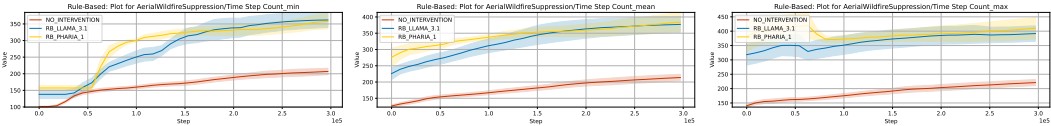

Figure 33: Time Step Count (**Rule-Based**) - No controller baseline VS Rule-Based Controller with Llama-3.1-8B Instruct: min, mean and max.

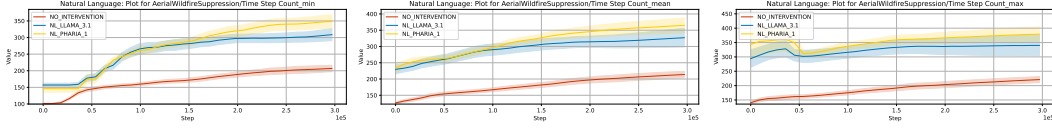

Figure 34: Time Step Count (**Natural Language**) - No controller baseline VS Natural Language Controller with Llama-3.1-8B Instruct: min, mean and max.

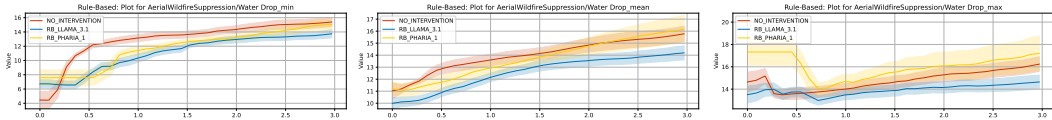

Figure 35: Water Drop Count **(Rule-Based)** - No controller baseline VS Rule-Based Controller with Llama-3.1-8B Instruct: min, mean and max.

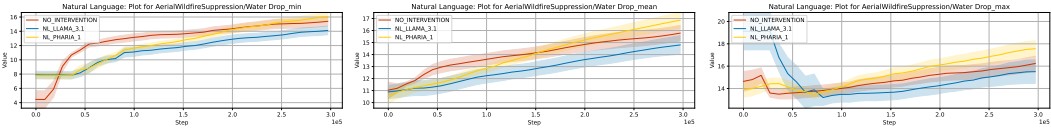

Figure 36: Water Drop Count **(Natural Language)** - No controller baseline VS Natural Language Controller with Llama-3.1-8B Instruct: min, mean and max.

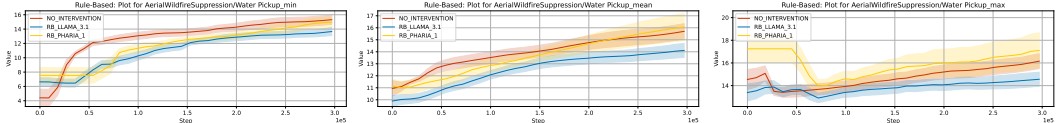

Figure 37: Water Pickup Count **(Rule-Based)** - No controller baseline VS Rule-Based Controller with Llama-3.1-8B Instruct: min, mean and max.

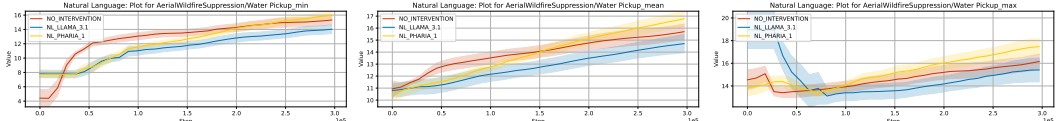

Figure 38: Water Pickup Count **(Natural Language)** - No controller baseline VS Natural Language Controller with Llama-3.1-8B Instruct: min, mean and max.

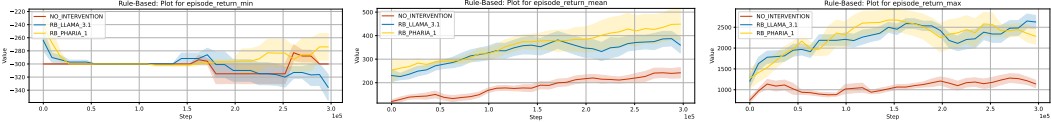

Figure 39: Episode Return **(Rule-Based)** - No controller baseline VS Rule-Based Controller with Llama-3.1-8B Instruct: min, mean and max.

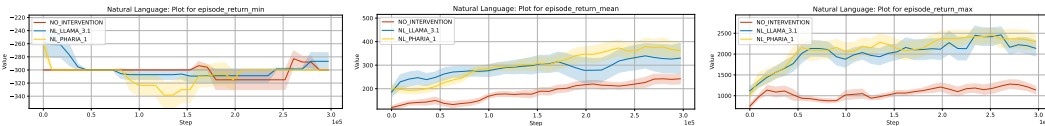

Figure 40: Episode Return **(Natural Language)** - No controller baseline VS Natural Language Controller with Llama-3.1-8B Instruct: min, mean and max.

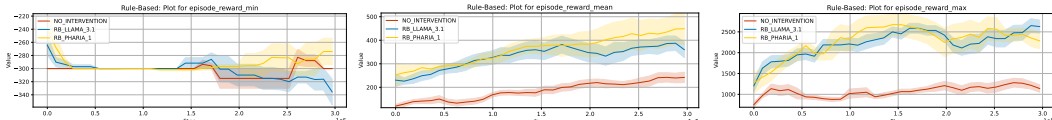

Figure 41: Episode Reward (**Rule-Based**) - No controller baseline VS Rule-Based Controller with Llama-3.1-8B Instruct: min, mean and max.

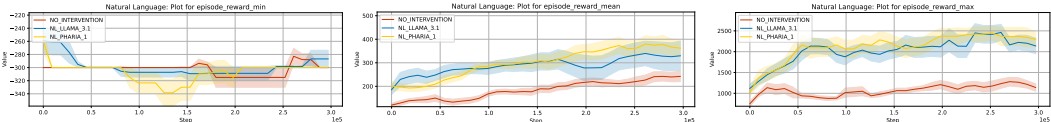

Figure 42: Episode Reward (**Natural Language**) - No controller baseline VS Natural Language Controller with Llama-3.1-8B Instruct: min, mean and max.

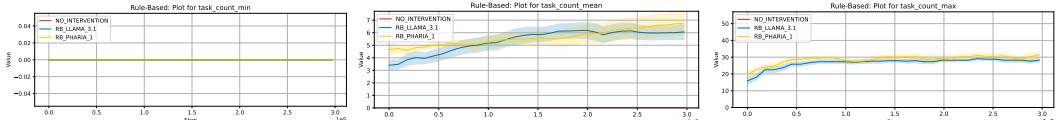

Figure 43: Task Count (**Rule-Based**) - No controller baseline VS Rule-Based Controller with Llama-3.1-8B Instruct: min, mean and max.

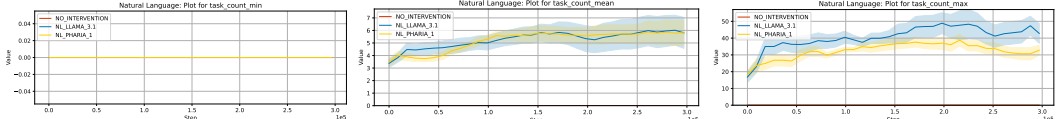

Figure 44: Task Count (**Natural Language**) - No controller baseline VS Natural Language Controller with Llama-3.1-8B Instruct: min, mean and max.

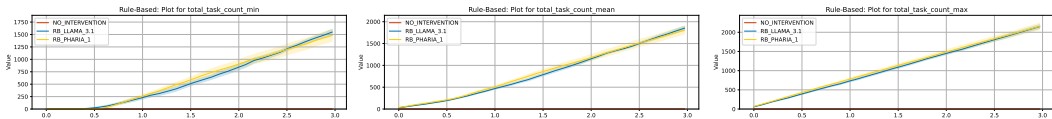

Figure 45: Total Task Count (**Rule-Based**) - No controller baseline VS Rule-Based Controller with Llama-3.1-8B Instruct: min, mean and max.

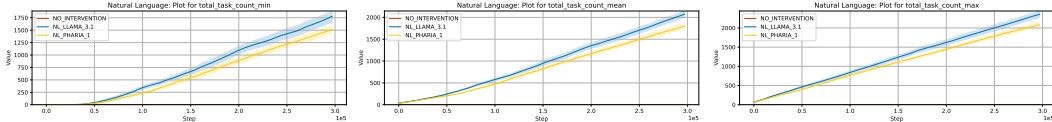

Figure 46: Total Task Count (**Natural Language**) - No controller baseline VS Natural Language Controller with Llama-3.1-8B Instruct: min, mean and max.

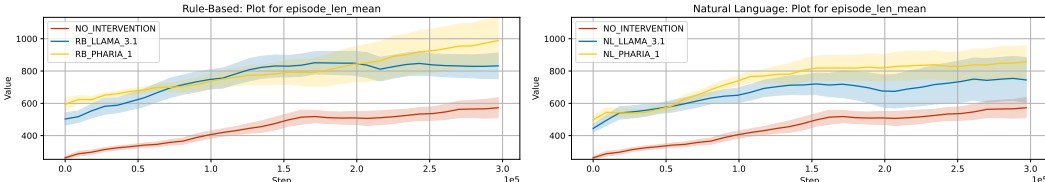

Figure 47: Episode Length - No controller baseline VS Rule-Based (left) and Natural Language (right) Controller with Llama-3.1-8B Instruct: min, mean and max.

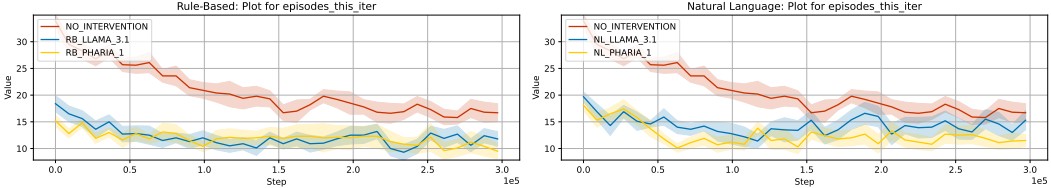

Figure 48: Episodes This Iteration - No controller baseline VS Rule-Based (left) and Natural Language (right) Controller with Llama-3.1-8B Instruct: min, mean and max.

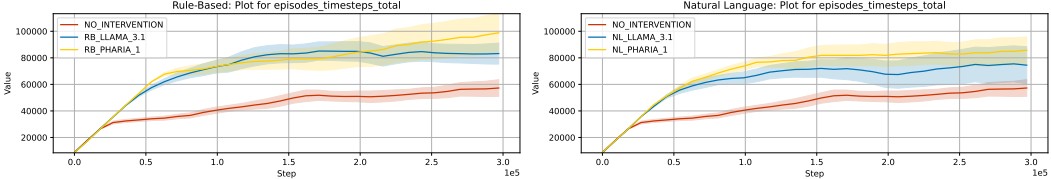

Figure 49: Episodes Timesteps Total - No controller baseline VS Rule-Based (left) and Natural Language (right) Controller with Llama-3.1-8B Instruct: min, mean and max.

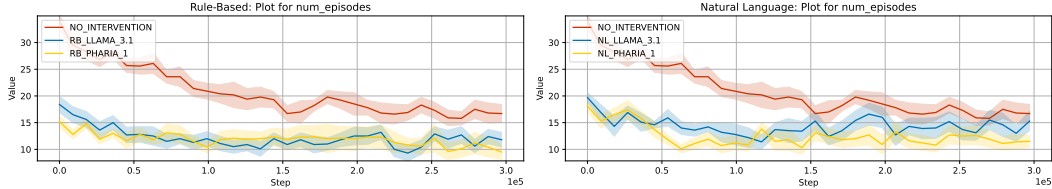

Figure 50: Number Episodes - No controller baseline VS Rule-Based (left) and Natural Language (right) Controller with Llama-3.1-8B Instruct: min, mean and max.

