# OpenReview forum: "LLM-Mediated Guidance of MARL Systems"
_ICLR.cc/2025/Conference — Submitted to ICLR 2025_

### Official Review · Reviewer_W4b8 · 2024-10-27

**Soundness:** 1
**Presentation:** 3
**Contribution:** 2
**Rating:** 3
**Confidence:** 4

**Summary:**

This paper investigates a method to use a pretrained LLM as a central controller, to guide otherwise decentralized MARL-trained agents. The environment is a simulation in which three drones have to contain wildfires, with observations consisting of a feature vector and a low-resolution RGB rendering of the drone's surroundings. The mediation approach is fairly simple, with either "soft-coded" (parametrized) rules, or natural language strategies being converted into actions, which individual agents are then forced to take. The paper's central claim is that this simple approach can significantly improve the agents' performance on complex tasks.

**Strengths:**

The paper is generally well-written and understandable. The overall idea of using an LLM to facilitate coordination is very interesting and likely to be valuable in the near future, as foundation models become more and more widespread and capable.

**Weaknesses:**

My main issue is the experimental validation of the approach, seeing as this is an entirely empirical paper with no significant theory.
In short, in the current state of the paper, I'm largely unconvinced that this method actually does provide significant benefits, and if so, how large they are.
This is largely based on the results presented in Figures 9 and 10:
1. The graphs are somewhat under-explained. They have very large shading, which is not explained in the paper - I'd assume it's a measure of dispersion, in this case, probably the full range of values between the 10 averaged runs (though it might be standard deviation/error?). There seem to be frequent outliers, there's significant overlap between all lines, and the upper bounds (and in some cases - the averaged (?) lines) are oddly non-monotonic. For example, there seems to be at least a single run with Llama 3.1 intervention at the very beginning of the training, that outperforms the final average performance. This might point at insufficiently tuned hyperparameters, unstable performance, or even an insufficient sample size.
2. The baseline seems undertuned. The environment seems to be relatively new, without a well-established baseline performance, so it's hard to tell whether it's really that difficult for regular PPO, or if better hyperparameters could significantly change the shape of the graphs.
3. The training doesn't seem to have converged - the performance of each method keeps increasing on average, although this may very well be significantly impacted by the statistical noise, so it's rather hard to tell.
4. Rule-based performance seems to largely match the natural language mediator - we see that also in Table 1, which also clearly shows the massive scale of the uncertainty, with uncertainty values often exceeding the averages. The role of the LLM in the rule-based system seems to be largely insignificant - it is merely converting structured data into natural language, for it to be converted back into structured data by the LLM controller. This is somewhat supported by the fact that RB performances for Llama 3.1 and Pharia 1 are very close. So if the results we get are very close between the rule-based system, and the LLM-based natural language system, why bother with the LLMs? We can see LLMs as a proxy for a human operator, but then we're confounding the quality of the LLM's strategy and the usefulness of the extra centralized information itself, plus the transcription fidelity of the controller.


Even if we accept that RB/NL interventions improve the performance over the baseline (which I can still accept on an intuitive level), the comparison hardly seems fair - as far as I understand, no-intervention models have no access to any global information, so they simply do not know where the fire might be if it's outside of their field of vision. Merely pointing them in the right direction would likely boost their performance significantly, and this information can be conveyed through the RB/NL systems.


There is also the question of how interventions affect the training process. As PPO is a (mostly) on-policy algorithm, training it on actions that have been taken due to the mediator override breaks all mathematical validity of the algorithm, at least without some form of e.g. importance sampling. I don't see any discussion of this in the paper, suggesting that those samples were just thrown into the rollout buffer without extra consideration. It is possible that training was performed without interventions, and performances are reported based on separate evaluation rollouts, but this should be stated explicitly in the paper.


Finally, I would really like to see some expreriments



Nitpicks:
- Lines 202-203 - incorrectly formatted citations
- Figure 3 describes both the environment logic ("drop water") and the algorithm itself (controllers), which are (or should be) entirely orthogonal. The algorithm does get its own diagram in Figure 4, but by the time this reader reached it, they already spent some brainpower trying to disentangle Figure 3.
- To get a sense of the complexity of the training, I'd be curious to see the actual wall time duration of the training. Of course this measure depends on many factors, but it's still good to know if it's minutes, hours, or days (or more?)

**Questions:**

1. What was your approach to hyperparameter tuning, especially for the baseline method (no intervention)?
2. Do you believe it is the case that regular RL algorithms largely fail on this task, as suggested by Figure 9?
3. Did you try running the baseline for a longer time to see whether the final performance here is stable?
4. Do you have any comments on the additional global information given to the RB/NL variants, which is inaccessible to the no-intervention variant?
5. How do you interpret the large uncertainties present in the training curves?

---

> ### Author Response · Authors · 2024-11-19
> **Weaknesses Comments**
>
> We sincerely appreciate your time and effort in reviewing our paper and mentioning that it is well-written and understandable. Further, thank you for acknowledging that our work is “very interesting”.
>
> **Weaknesses: My main issue is the experimental validation…**
>
> Answer: We have re-run all experiments due to a bug when syncing policy action requests and frame-rates of the environment. The results show more significant improvement on “extinguished burning trees” and “episode reward mean”. While our method shows a 2x improvement over the non-intervention baseline on “episode reward mean”, the improvement on “extinguished burning trees” is roughly 15x.
>
> New results:
>
> “Table: No controller, RB and NL Controller performance on *Episode Reward Mean¹* and *Extinguishing Trees Reward²* for Llama-3.1-8B Instruct and Pharia-1-LLM-control-aligned.”
>
> | Mediator       | Size | Controller        | Episode R. Mean¹         | Ext. Trees R.²            |
> |----------------|------|-------------------|--------------------------|---------------------------|
> |                |      | None              | 238.34 (±14.34)          | 1.18 (±0.16)              |
> | Pharia-1-LLM   | 7B   | Rule-Based        | **437.65** (±43.28)      | 13.75 (±1.38)             |
> |                |      | Natural Language  | 372.05 (±24.45)          | 5.89 (±0.79)              |
> | Llama-3.1      | 8B   | Rule-Based        | 376.18 (±21.98)          | **15.76** (±1.76)         |
> |                |      | Natural Language  | 331.22 (±39.88)          | 6.73 (±0.81)              |
>
>
>
> 1. Answer: Instead of reporting our results using min, mean and max area plots we have decided to change this to error of the means, which make the plots much better readable and method improvements are more comprehensible.
>
> 2. Answer: Good point, however is this impression based on no existing results / experiments on hyperparameter sweeps? If so, this might be the result of limited compute resources.
> 3. x
> 4. Answer: New results post bug-fix do not show that “performances for Llama 3.1 and Pharia 1 are very close”. No “extra centralised information” exists for any rule-based and no-intervention experiments. The only additional information the natural language-based experiment setup has is the static location of the village. However, this information does not help improve the performance of the NL-based intervention, which is expected as it is harder for the LLM(s) to understand free-form natural language than structured. 4. a) “This is somewhat supported by the fact that RB performances for Llama 3.1 and Pharia 1 are very close.”
> Answer: This does not hold true anymore with new results. 4. b) “So if the results we get are very close between the rule-based system, and the LLM-based natural language system, why bother with the LLMs?” Answer: The “LLMs” imitate a human user
>
>
> **Weaknesses: Even if we accept that RB/NL interventions improve the performance…**
>
> Answer: The only global information that is not provided directly to the RB and No-Intervention experiments is the static location of the village. The main comparison is between no intervention and two types of interventions, structured and unstructured (free-form natural language). While the unstrctured- intervention (NL) has some “extra” information it is still surpassed by the RB experiment setup. This still proofs that the method works very well, but with structured input.
>
> **Weaknesses: Nitpicks:**
>
> “Lines 202-203 - incorrectly formatted citations.” Answer: Fixed.
>
> “Figure 3 describes both the environment logic ("drop water") and the algorithm itself (controllers), which are (or should be) entirely orthogonal. The algorithm does get its own diagram in Figure 4, but by the time this reader reached it, they already spent some brainpower trying to disentangle Figure 3.” Answer: Answer: Fixed order, thank you for pointing this out.
>
> “To get a sense of the complexity of the training, I'd be curious to see the actual wall time duration of the training. Of course this measure depends on many factors, but it's still good to know if it's minutes, hours, or days (or more?).“ Answer: Thank you for pointing this out, we will add this information.

---

> ### Author Response · Authors · 2024-11-19
> **Question Answers**
>
> **Questions:**
>
> 1. Answer: There were no hyperparameter sweeps due to limited compute resources. The results for this paper have been produced on a single 3090. We believe this should not undermine the validity for the method as the hyperparameters for the RL setup in all three experiments are the same and hence we believe compareable. Even if the hyperparameters could be considered “sub-optimal” our results actually raises a point / question towards defeating the need for large hyperparameter sweeps when using our method (we could discuss this further in discussion section).
>
> 2. The new results do not suggest that, and we don’t believe that this is the case.
>
> 3. We have run this for a longer time and the baselines seem to be more stable, however, we believe early stopping can convey the potential of our method.
>
> 4. The only additional global information as part of the “system prompt” is the static location of the village. This information is given to the NL experiment setup.
>
> 5. This has been eliminated as “error of the means” differentiating the performance more clearly and produces more legible plots.

---

> > ### Author Response · Authors · 2024-11-22
> >
> > We would like to kindly notify you that we have uploaded a Rebuttal Revision considering and incorporating your feedback, possible in the discussion period time-frame.
> >
> > **Major**
> >
> > We believe the new results table as well as additional results in the Appendix A.6 Additional Results might alleviate some of your main concerns such as **Weakness 4**, **Question 2** and **5**.
> >
> > **Minor**
> >
> > We appreciate your attention to detail and are grateful for your "nitpicks"—details do matter. In response, we have corrected the formatting in lines 202–203 and added an annotation to Figure 3 that references Figure 4. This allows readers to directly consult Figure 4 for a more detailed understanding of the LLM-Mediator. While we considered swapping the order of these two figures, we ultimately decided against it. Replacing Figure 3 with Figure 4 might confuse readers, as it could imply that the LLM-Mediator fully replaces the actions of the RL agents, rather than temporarily overriding them.
> >
> > Thank you for your continued consideration and time spent, much appreciated.

---

> > > ### Author Response · Authors · 2024-11-26
> > >
> > > Dear Reviewer W4b8, we hope this message finds you well. We would be deeply grateful for your thoughts on the responses we have provided and whether there is anything further we could clarify or elaborate on. Our aim is to refine our work to the fullest extent by thoughtfully incorporating your valuable feedback and identifying clear, actionable steps. Any additional insights you could share would be immensely appreciated. Thank you

---

### Official Review · Reviewer_eHD7 · 2024-10-28

**Soundness:** 3
**Presentation:** 4
**Contribution:** 3
**Rating:** 6
**Confidence:** 5

**Summary:**

This paper investigates how LLMs can be used to interpret and facilitate interventions that shape the learning trajectories of multiple agents. The experiments including two types of interventions, referred to as controllers: a Natural Language (NL) Controller and a Rule-Based (RB) Controller. The NL Controller, which uses an LLM to simulate human-like interventions, showed a stronger impact than the RB Controller. This paper indicates that agents particularly benefit from early interventions, leading to more efficient training and higher performance. The experiments were designed to compare agents’ performance under three different intervention setups: No Controller, RB and NL Controller. Both intervention types outperform the baseline without interventions, highlighting the potential of LLM-mediated guidance to accelerate training and enhance MARL performance in challenging environments.

**Strengths:**

1, This approach moves beyond static curriculum-based methods, providing real-time, adaptive interventions that respond to the evolving states of agents and environments.

2, The results demonstrate that interventions, especially during early training, accelerate learning to reach expert-level performance more efficiently.

3, The authors provide a thorough comparison between different types of controllers (RB, NL, and no intervention), making it clear that LLM-mediated interventions provide substantial performance improvements. And also this paper shows the different results by using different LLMs, which indicates that this method can be utilized with different LLMs.

**Weaknesses:**

1,Scalability: The experiments are all conducted using the default settings of 3 agents. Given that the paper aims to address multi-agent scenarios, it would be more convincing to demonstrate the method's performance with a larger number of agents, such as 4, 5, or 6 agents, to showcase its scalability more effectively.

2,MARL Method: The experiments exclusively use Proximal Policy Optimization (PPO) as the base MARL algorithm. To strengthen the findings, it would be beneficial for the authors to implement and evaluate the method using other MARL algorithms like QMIX, VDN, MADDPG or other MARL methods, providing a broader assessment of its effectiveness.

3,Test Environment: The testing is limited to the Aerial Wildfire Suppression (AWS) environment. There is no evaluation demonstrating the generalizability of this method to other environments, such as StarCraft II, Hanabi, or other established multi-agent benchmarks. Expanding the experiments to include these environments would provide stronger evidence of the method's versatility.

**Questions:**

The method is clear and it does make sense to me. Like I state in the weaknesses, if there would be more results to prove the performance of this method, it would be more convincible.

Moreover, there are a lots of works of using LLM to do reward generation to help reinforcement learning training. What is the advantage of proposed method compared to reward generation by LLMs? Such as Text2reward, DrEureka, Reward design with language models.

---

> ### Author Response · Authors · 2024-11-19
>
> Thank you for your thoughtful review and careful reading of our paper. We appreciate the recognition of our adaptive approach that moves beyond static methods. Thank you for noting the efficiency of early interventions in accelerating learning. We are also grateful for your acknowledgment of our comparisons, which show clear performance improvements across controllers and LLMs.
>
> **Weaknesses**
>
> 1. Answer: Thank you for pointing this out. We will run and add experiments on this with 6 and 9 agents.
> 2. Answer: Due to limited compute resources, this is difficult to execute on.
> 3. Answer: Fair point; this could be done in future work.
>
> **Questions**
>
> “Moreover, there are a lot of works…”
>
> Answer: We believe we have contrasted this (LLM reward generation) question in our related work, but we will extend on this, thank you for pointing this out.

---

> > ### Comment · Reviewer_eHD7 · 2024-11-19
> >
> > So I changed my rating from 5 to 6. I know there might still be some limitations and weaknesses of this paper. But in general, I do believe that this is a promising direction so I wish there would be more people to see this work since this is a solid work. I hope that author could keep refining this work and I am also excited about how this method works in other environment and using other MARL baseline. And I am interested to see how this method actually work after you open source it so the community could play around your code and test by ourselves in the future. Wish you all the best and good luck. Thanks for your reply and hard working.

---

> > > ### Author Response · Authors · 2024-11-19
> > > **Thank you!**
> > >
> > > Thank you very much for your thoughtful feedback, efforts, time spent and for increasing your rating. We appreciate your recognition of the potential and effort behind this work. Your encouragement motivates us to keep refining and expanding on this method. We’re excited to explore its applications in other environments and MARL baselines, and we’re committed to keeping open-sourcing all code to enable the community to experiment with it further. Thank you again for your support and valuable insights.

---

> > > > ### Author Response · Authors · 2024-11-22
> > > >
> > > > Dear Reviewer eHD7, we would like to kindly inform you that we have decided to look at **Weakness 1,Scalability** and are running "4, 5, or 6 agents" count ablations and hope to report results soon.

---

> ### Author Response · Authors · 2024-11-26
>
> We are pleased to inform you that we have updated our manuscript to include Rule-Based Intervention scalability experiments. These experiments test setups involving 3, 4, 5, and 6 agents using Pharia-1 and Llama 3.1 agents versus no intervention baseline. The updates have been incorporated into the Results section, with additional discussion added to the Discussions section.
>
> Thank you for your time and effort and considering our changes.

---

### Official Review · Reviewer_rf94 · 2024-10-30

**Soundness:** 2
**Presentation:** 3
**Contribution:** 2
**Rating:** 6
**Confidence:** 4

**Summary:**

This empirical study paper investigated how LLMs can be used as a controller to intervene the learning process of MARL. The main contributions of this paper are evaluating the capability of two representation of prompts, referred to as rule-based controller and natural language controller. The later one can give some insights into the potential performance of human-in-the-loop training for MARL. The experments were conducted on a complex environment called Aerial Wildfire Suppression, and showcased the effectiveness of intervened MARL learning process by LLMs.

**Strengths:**

1. From the perspective of an empirical and engineering work, this paper seems like an original work, as I cannot find duplicated counterparts online. Although the paradigm of intervening MARL learning process is not a completely new idea, the leverage of LLMs to produce the interventions should be novel.
2. The motivation of this paper is clear, and the descriptions of appraoches and experimental platforms are comprehensive. This paper well took the advantage of one additional page to provide illustrative figures, which can aid in understanding the paper.
3. The overall quality of this paper is good, which is evident in the complete discussion of limitations of the proposed paradigm and its gap to the realistic applications. The general style of this paper should lie in an empirical study paper, based on which I think it is an examplar that include every component it should involve. Furthermore, this paper has released the code, and the details of experimental settings is evident to reproduce the results.
4. As the prevalence of LLMs, the research trend and flavour in many fields have been influenced, which is an inreversible situation. To adapt to this tendancy, the combination of LLMs and MARL is reasonable, if this can really help improve the performance or close the gap to the realistic application. The trial made in this paper well shedded light on the possibility of this combination, which is thrilled to MARL reseachers such as myself. In this sense, I believe this paper studied significant work.

**Weaknesses:**

1. This experiments lack some instrucmental baselines to verify the importance of LLMs to generate interventions, such as the baseline which replaces the LLMs by a reinforcement learning agent (learner). I believe with this comparison, the result of this paper can be more convincing.
2. As for the statement in line 499-501 that "The findings suggest ... in complex environments," I wonder if this claim is certainly correct, as the computational cost from LLMs is known to be expensive also. If the cost induced by LLMs can offset the cost of extra costs led by explorations is deserved to investigate further.
3. Although this paper have made a good description about its approach and experimental results, the insightful analysis is lacking. For example, to most of MARL researchers, the most prominent question could be what mechanism the LLM-based controller has learned. It is better if this paper can demonstrate and analyze some intervention strategies learned, other than the numeric results.
4. As for a paper to be published on top venues, I believe the demonstration on the only one setting as extinguishing burning trees is not enough to convince others that the proposed approach is effective. To alleviate this weakness, I suggest you may do more experiments on other settings with different objectives.

**Questions:**

Please see weaknesses and give a reply to those concerns.

---

> ### Author Response · Authors · 2024-11-19
>
> Thank you for your thoughtful review, careful reading of our paper and for recognizing the originality of our work and the novelty of leveraging LLMs for MARL interventions. We appreciate your acknowledgment of our motivation, comprehensive descriptions, and use of illustrative figures to enhance understanding. We are also grateful for your recognition of our discussion on limitations and the reproducibility of our results through shared code. Lastly, we are encouraged by your support for the combination of LLMs and MARL and its potential to inspire future research in this field. Thank you for highlighting the significance of our study.
>
> **Weaknesses**
>
> 1. Answer: The non-intervention baseline is a “pure” RL learner using PPO.
> 2. Answer: Fair point.
> 3. Answer: The point of the mediator guidance could be seen as a real-time dynamic LLM-based immitation learning mechanism. This system is not about learning to understand strategies yielded by the LLM. It is about pushing the RL learning into the right direction, which is especially helpful assuming NN weights of the RL agent are initialized at random.
> 4. Answer: We will add more metrics, apart from extiguishing trees and average cumulative reward, could also show, water-dropt, water-pickup, prepare trees etc. in the main body of the paper, however we don’t believe that this would help to further support the performance increase of our method.

---

> > ### Comment · Reviewer_rf94 · 2024-11-21
> >
> > Thanks for your reply.
> >
> > For question 4, the main objective is to make the analysis as complete as possible from the systematic view. Different people may have different angle of ineterst on this system.
> >
> > For question 3, understanding is always critical if you would claim that your system is useful, as trustworthiness is prior to the capability of a system designed for real-world problems. This is the position of this paper in my mind, which is the reason why I give 5 before.

---

> > > ### Author Response · Authors · 2024-11-22
> > >
> > > Thank you very much for your considered comments and thoughtful review.
> > >
> > > **"For question 4, the main objective..."**
> > >
> > > Answer: Thank you, we will add all resulting data including plots.
> > >
> > > **"For question 3, understanding..."**
> > >
> > > Answer: The idea behind the environment is to move closer to real-world scenarios, beyond toy / game / grid-world environments, while be believe in incrementally closing the sim-to-real gap, we acknowledge that we are also far away from a realistic scenario, however the main focus of the paper really is supposed to be the LLM guided intervention to enhance the selected MARL system.
> > >
> > > We noticed that in your comment, you mentioned, "the reason why I give 5 before." We wanted to kindly confirm whether the rating of 5 reflects your final assessment, or if there are any remaining concerns or updates you'd like us to address further to refine the paper.

---

> > > > ### Comment · Reviewer_rf94 · 2024-11-22
> > > >
> > > > If you can give me some more results within the discussion period, I will consider to increase my score.

---

> > > > > ### Author Response · Authors · 2024-11-22
> > > > >
> > > > > Thank you very much for your time and effort and continued consideration of our work.
> > > > >
> > > > > We have uploaded a Rebuttal Revision and added all additional results that we have logged during training in the Appendix A.6 - **Additional Results**
> > > > >
> > > > > Please let us know if there is anything else we can provide to further clarify and strengthen our submission.

---

> > > > > > ### Comment · Reviewer_rf94 · 2024-11-22
> > > > > >
> > > > > > Thanks for your additional results. I will increase me score to 6.

---

> > > > > > > ### Author Response · Authors · 2024-11-22
> > > > > > > **Thank you!**
> > > > > > >
> > > > > > > Thank you sincerely for your valuable feedback, the time and effort you dedicated, and for raising your rating. We deeply appreciate your recognition of the effort and potential behind this work. Your encouragement inspires us to continue building upon it. Please let us know if you have any further suggestions that could help us improve this work.

---

### Official Review · Reviewer_5BcY · 2024-11-01

**Soundness:** 2
**Presentation:** 3
**Contribution:** 1
**Rating:** 3
**Confidence:** 4

**Summary:**

In this work, the authors propose introducing LLM-mediated interventions to support multi-agent reinforcement learning (MARL) systems. In particular, the authors propose two separate strategies (i.e., rule based, natural language) designed to map linguistic specifications to agent instructions. The rule based approach is implements a more rigid protocol (e.g., "go to the closest fire") while the natural language approach involves prompting an LLM to develop a more dynamic strategy. The authors propose a novel  Aerial Wildfire Suppression (AWS) MARL environment to benchmark their approaches and report experiments indicating that the proposed interventions (may) increase reward over a no controller baseline.

**Strengths:**

### Presentation
The authors provide a clear and polished description of their proposed system and the problem that it solves. I was able to easily follow the flow of ideas being presented throughout the work, and found the visual illustrations of the AWS system helpful.

### Motivation
The problem that the authors study --- using Natural Language Instructions to coordinate multi-agent systems --- is well motivated and compelling. Generally, there is an opportunity for natural language specifications to improve the adaptability of coordination of agents interacting in an environment.

### Replicability
The authors provide code for replicating their study design and experiments.

**Weaknesses:**

### Novelty
My primary concern with this work is that the novelty of the proposed framework is unclear.

The authors claim that no existing systems support the use of LLMs for MA learning systems (line 43). However, recent work has studied this problem (see Sun & Huang, 2024 and references therein for a recent overview). Additionally, while the authors claim that the proposed AWS environment has more complex environmental dynamics than prior work (i.e., C1 on line 110), it is unclear how the proposed benchmark differs substantively different dynamics from existing simulation environments such as Minecraft (e.g., Wang et al., 2023).

Further, the rule-based and natural language controller implementations, while useful points of comparison, appear to be engineering system implementation efforts as opposed to research contributions. The authors also mention this in the statement of contributions C1, explaining “We implement a novel *system*.”




### Experimental Evaluation
Based on the current experimental results, I am unable to make an assessment as to the proposed LLAMA/PHARMA interventions over the no intervention conditions. In particular, the method used to construct statistical uncertainty intervals is not reported, and the intervals overlap over time steps. Additionally, the current performance results do not indicate a robust reward trend over time steps.

On a more minor note, updating the coloring to shades that are easy to distinguish in Figures 9 and 10 would be helpful.

Furthermore, to further validate the proposed approach in the future (i.e., rule-based and natural language controller generated interventions), it would be helpful to provide a comparison against existing MARL approaches and compare across additional environments. Given the current results, it is unclear how well the provided results compare against other MARL frameworks or generalize to different environments.

Wang et al., Voyager: An Open-Ended Embodied Agent with Large Language Models (http://arxiv.org/abs/2305.16291)

Sun & Huang, LLM-based Multi-Agent Reinforcement Learning: Current and Future Directions (https://arxiv.org/pdf/2405.11106)

**Questions:**

1. How does the design of the proposed framework differ from prior MARL systems, particularly those that leverage natural language instructions?
2. How does the performance of the proposed NLI and Rule-Based controllers improve upon existing MARL approaches, particularly those cited in response to question one above?
3. What specific characteristics in the AWS evaluation system differ from prior MARL evaluation environments? Why might future work use the AWS framework over these alternatives?

---

> ### Author Response · Authors · 2024-11-19
>
> Thank you for dedicating your time to review and evaluate our paper. We sincerely thank the reviewer for their feedback. We are pleased that the clarity and flow of our presentation, along with the visual illustrations of the AWS environment and system, were helpful in conveying our ideas. We appreciate your recognition of the motivation behind using natural language instructions for multi-agent coordination and its potential to enhance adaptability. Lastly, we are glad that our efforts to ensure replicability through shared code were acknowledged. Your comments validate the key aspects of our work and its broader relevance.
>
> **Weaknesses: Novelty**
>
> “The authors claim that no existing systems…”
>
> Answer: Minecraft, to our knowledge, is not an MA environment (MineRL).
>
> “Further, the rule-based and natural language controller implementations…”
>
> Answer: The contribution is the environment as well as the LLM mediated guidance system using a natural language and rule-based system.
>
> **Weaknesses: Experimental Evaluation**
>
> “On a more minor note, updating the coloring…”
>
> Answer: Fixing the color selection, thank you for pointing this out.
>
> “Furthermore, to further validate the proposed approach in the future…”
>
> Answer: The point is not to compare against other MARL frameworks, it is to demonstrate how the proposed system can enhance existing MARL systems. The comment about generalizing to different environments is a valid point. The following References will be incorporated in our related work section.
>
> **Questions**
>
> 1. Answer: Thank you for pointing out the works by Sun & Huang and Wang et al., we greatly appreciate you bringing them to our attention. We will gladly incorporate these references into our work and provide a detailed contrast to highlight the distinctions and contributions of our approach.
>
> 2. Answer: Considering the additional references you mentioned, our method does not fall neatly into either the Open-loop or Closed-loop categories as defined in Sun & Huang. For instance, one defining characteristic of an Open-loop system is that "the reward...is not considered" (Sun & Huang). In our case, however, our approach also doesn’t align with Closed-loop systems, as we temporarily replace RL agent actions with those guided by the LLM intervention in both NL and RB setups. Furthermore, while Sun & Huang discuss MARL systems that leverage LLMs for communication and collaboration (e.g., through memory), our approach uniquely focuses on using a central LLM to craft strategies to guide multiple agents simultaneously. To the best of our knowledge, this aspect remains unexplored and represents a key contribution of our work.
> Additionally, we believe our approach aligns with the direction outlined in "IV. OPEN RESEARCH PROBLEMS" – specifically "B. Language-enabled Human-in/on-the-Loop Frameworks" (Sun & Huang), as we effectively mimic human-in-the-loop strategies using LLMs. In contrast to Wang et al. ("Voyager"), which focuses on building abilities with LLMs, our work develops centralized, high-level strategies to guide multiple agents. We believe this distinction highlights a novel and important aspect of our research.
>
> 3. Answer: Addressing this question fully would require a significant number of additional experiments. While it’s an intriguing avenue for future work, we appreciate your understanding that it goes beyond the current scope of our study.

---

> ### Author Response · Authors · 2024-11-22
>
> We want to kindly notify you that we have uploaded a Rebuttal Revision.
>
> We have identified and incorporated the following action items:
> - Contrast and include the suggested references mentioned by you as part of our Related Work section
> Wang et al., Voyager: An Open-Ended Embodied Agent with Large Language Models (http://arxiv.org/abs/2305.16291)
> Sun & Huang, LLM-based Multi-Agent Reinforcement Learning: Current and Future Directions (https://arxiv.org/pdf/2405.11106)
> - We have updated our colouring scheme to improve the legibility of our training result graphs.
>
> Additionally, we have updated our results table, including wall-time, and added additional results in Appendix A.6 Additional Results.
>
> Thank you for your time and additional consideration.

---

> ### Comment · Reviewer_5BcY · 2024-11-25
>
> I would like to thank the authors for their detailed rebuttal -- I have reviewed the comments and changes to the paper. However, my concerns continue to stand and I would like to retain my score.

---

### Official Review · Reviewer_XdBm · 2024-11-04

**Soundness:** 1
**Presentation:** 3
**Contribution:** 1
**Rating:** 3
**Confidence:** 4

**Summary:**

This article presents a method for solving MARL tasks by inserting vectorized observations from the environment into an LLM prompt template and obtaining decisions from LLM mediators in two different ways (RB and NL). The aim is to leverage the reasoning and planning abilities of LLMs to assist in completing the fire extinguishing task in an environment called Aerial Wildfire Suppression.

In the rule-based method, the LLM mediator takes a prompt encoded with environment observation vector as input and outputs high-level actions for the environment. While, the natural language method allows the LLM mediator to output high-level actions in the same format by taking a natural-language strategy generated from the Strategy LLM, which uses an input prompt containing partial observations.

**Strengths:**

This article claims that the novelty lies in using LLM as a mediator to intervene between the action policy and the environment, dynamically. Especially, in the Natural-language intervention method, the prompt-encoded environment features are fed into a Strategy LLM and then the natural language outputs are fed into the LLM-Mediator for real-time action policy.

The writing style made this article easy to read and understand.

**Weaknesses:**

1. **Lack of novelty.**

As mentioned in strengths, what this article claims as novelty is not quite robust. The novelty from my side is: translating prompt to natural language strategy and then translate it back to formatted expression. Embedding the environment information into LLM prompt has been studied by a lot of works in different ways. Simply applying this idea to multi-agent environment, as one of the "centralized training and execution (CTE)" methods, does not contribute to this area.

2. **The technical part is not detailed enough.** Besides the figures and the corresponding demonstration in Section4, it seems nothing else.

a) How does this method train policy with MARL using the two LLMs(Pharia-1-LLM-7B-control-aligned and Llama-3.1-8B Instruct)? How do you collect the data for MARL? How is the learned RL policy used through this LLM-Mediator framewrok?

b) Why does this article choose CTE but not others, like "centralized training and de-centralized execution (CTDE)"? Any insight behind the choice?

c) In which part of multi-agent envrionment does this method improve? Do the agents collaborate better with each other?

3. **Evaluation part is short and not convincing.**

a) From Figure9 and Figure10, the performance of RB and NL does not seem significantly better than what this article claimed at line428 "significantly enhance agents’ performance in complex environments like AWS".

b) Only one benchmark may not be enough. The authors should show the robustness for the method among different kinds of environments.

c) Selecting 'No Intervention' as the only baseline is not convincing.

4. **This article does not allocate a reasonable length to each section.**

There are too much introduction to the environment AWS and the discussion of "LIMITATIONS AND POTENTIAL IMPACTS", making sections 4, 5, and 6 too short for detailed demonstration.

**Questions:**

See above in "Weaknesses" part.

---

> ### Author Response · Authors · 2024-11-19
>
> Thank you for taking the time to read and review our paper and we were pleased to hear that it was easy to read and understand.
>
> **Weakness: 2. The technical part is not detailed enough.**
>
> a) Answer: The LLM-based actions are injected at specified time intervals, overwriting the early potentially sub-optimal actions that are predicted by the RL trained neural network.
>
> b) Answer: We are doing CTDE and will clarify that in the manuscript; thank you for pointing this out.
>
> c) Answer: The agents have no direct way in collaborating, however this is possible to emerge as they have a shared memory of the “last seen burning tree”. The LLM mediator however, can assign roles / tasks to the different agents, guiding agents in achieving collective goals.
>
> **Weakness: 3. Evaluation part is short and not convincing.**
>
> a) Answer: We have discovered a bug where the action and fire-growth steps were not synced up because of frame-rate dependencies. We have re-run all experiments and report new results that support that claim more significantly:
>
> "Table: No controller, RB and NL Controller performance on *Episode Reward Mean¹* and *Extinguishing Trees Reward²* for Llama-3.1-8B Instruct and Pharia-1-LLM-control-aligned."
>
> | Mediator       | Size | Controller        | Episode R. Mean¹         | Ext. Trees R.²            |
> |----------------|------|-------------------|--------------------------|---------------------------|
> |                |      | None              | 238.34 (±14.34)          | 1.18 (±0.16)              |
> | Pharia-1-LLM   | 7B   | Rule-Based        | **437.65** (±43.28)      | 13.75 (±1.38)             |
> |                |      | Natural Language  | 372.05 (±24.45)          | 5.89 (±0.79)              |
> | Llama-3.1      | 8B   | Rule-Based        | 376.18 (±21.98)          | **15.76** (±1.76)         |
> |                |      | Natural Language  | 331.22 (±39.88)          | 6.73 (±0.81)              |
>
>
> b) Answer: The environment has a large observation space of visual and vector observations and continuous and deterministic actions, we believe this could scale to simpler environments.
>
> c) Answer: Would you be so kind as to be specific here? What other baseline should be added?

---

> > ### Author Response · Authors · 2024-11-22
> >
> > Dear Reviewer XdBm, we would like to kindly inform you that we have uploaded a Rebuttal Revision, which includes additional details on our results in the main body of the paper under the Results section, as well as in Appendix Section A.6: Additional Results. We hope these updates address some of your concerns regarding performance and the perceived lack of improvement compared to the baseline, particularly the issue you highlighted as **Weakness 3a)**.
> >
> > Thank you once again for your time and thoughtful review of our work. Please don’t hesitate to let us know if there are additional changes we could make to further strengthen our submission.

---

> > > ### Author Response · Authors · 2024-11-26
> > >
> > > Dear Reviewer XdBm, we would greatly appreciate your thoughts on the responses we have provided and whether there is anything further we can clarify or expand upon. Our goal is to enhance our work as much as possible by incorporating your valuable feedback and identifying actionable steps. Any additional insights you could share would mean a great deal to us. Thank you again for your time and support.

---

> > ### Comment · Reviewer_XdBm · 2024-11-27
> >
> > Thanks for addressing some of my questions. But I still have concerns:
> > 1. The MARL training process should be displayed in the paper. It is better to have a pseudo code algorithm to demonstrate: 1) how the training data for RL is collected? Since it is claimed "CTDE", how the feature vectors and the visual observations from different agents are combined to train the policy? 2) After controllers craft intervention prompt, LLM-Interpreter(LLM-Mediator) assign task to agents and then when LLM-based actions are injected to replace RL-based actions?
> > I think this part should not be missing from an RL paper, not just the PPO pseudo code shown in the appendix.
> > 2. It seems reward value defined for each event in the environment shown in Appendix A.5 includes highly human intention. Could you explain why the value are choosen? I have the concern that the choice of value might influence the evaluation of performance. For example, the "Crossed Border" and "Too Close to Village" event is given very high penalty (-100 and -50), the "Episode R. Mean" could be improved a lot by maybe only avoid the happening of these two events. It is better to do fine-grained analysis from which event the improved part of reward value
> > mainly come from.
> > 3. The environment choosed by this paper have 8-dim feature vector and 42*42 visual observation space and it is claimed the environment is complex enough. I might be curious about the performance of this method on some house-holding environments, for example, ThreeDWorld Transport Challenge[1], which has much complex observation space (256*256 visual observation and more than 8-dim feature vectors).
> >
> > [1] Chuang Gan et al., The ThreeDWorld Transport Challenge: A Visually Guided Task-and-Motion Planning Benchmark for Physically Realistic Embodied AI, 2021

---

> ### Author Response · Authors · 2024-11-27
>
> Dear Reviewer XdBm, thank you very much once again for your thoughtful feedback.
>
> 1. The MARL training process should be displayed in the paper. It is better to have a pseudo code algorithm to demonstrate: 1) how the training data for RL is collected? Since it is claimed "CTDE", how the feature vectors and the visual observations from different agents are combined to train the policy? 2) After controllers craft intervention prompt, LLM-Interpreter(LLM-Mediator) assign task to agents and then when LLM-based actions are injected to replace RL-based actions? I think this part should not be missing from an RL paper, not just the PPO pseudo code shown in the appendix.
> > We greatly appreciate your feedback and have incorporated your suggestion by adding a new section to the main body of our paper: Pseudocode: MARL with LLM Intervention. This section provides detailed pseudocode explaining how our method works. Additionally, we refer to complete code of our implementation, which is available using RLib for further details. These changes have been included in the updated rebuttal submission.
>
> 2. It seems reward value defined for each event in the environment shown in Appendix A.5 includes highly human intention. Could you explain why the value are choosen? I have the concern that the choice of value might influence the evaluation of performance. For example, the "Crossed Border" and "Too Close to Village" event is given very high penalty (-100 and -50), the "Episode R. Mean" could be improved a lot by maybe only avoid the happening of these two events. It is better to do fine-grained analysis from which event the improved part of reward value mainly come from.
> > We acknowledge that this appears as human intention; however, we have tested this thoroughly. Designing and testing various reward setups has been complex and somewhat unstructured, as it often can be in RL setups. However, our primary objective was to establish a reward setup that enables fair comparisons across all experiments. To this end, the reward distribution outlined in Appendix A.5 supports the RL agents in all experiment setups (with or without intervention) in maximizing the primary metric of the Aerial Wildfire Suppression environment: the number of extinguished trees/reward. Also worth mentioning at this point for further investigation: All statistics, including isolated and mean rewards, including metrics such as extinguishing tree count, crashing / crossing border rewards have been reported in Appendix A.7 Additional Results.
>
> 3. The environment choosed by this paper have 8-dim feature vector and 4242 visual observation space and it is claimed the environment is complex enough. I might be curious about the performance of this method on some house-holding environments, for example, ThreeDWorld Transport Challenge[1], which has much complex observation space (256256 visual observation and more than 8-dim feature vectors).
> > Thank you for bringing up a fair point and an excellent reference. In future work, we aim to explore a broader range of environments, both existing and new, with varying levels of complexity.

---

### Meta-Review · Area_Chair_xtng · 2024-12-21

**Metareview:**

After careful consideration of the six expert reviews, the subsequent author-reviewer discussions, and the area chair's own reading of the paper, I recommend rejecting this submission. While the paper presents an interesting approach to incorporating LLM-mediated interventions in multi-agent reinforcement learning systems, several fundamental concerns prevent it from meeting ICLR's acceptance standards.

The paper's primary technical contribution centers on using LLMs as mediators to guide MARL agents through two types of interventions: rule-based and natural language controllers. While some reviewers acknowledged the potential value of this direction, particularly for future human-AI interaction scenarios, the technical execution and experimental validation fall short in several critical aspects.

A major concern shared across multiple reviewers is the limited scope of the experimental evaluation. The work relies solely on PPO as the base MARL algorithm without comparisons to more sophisticated state-of-the-art CTDE-based approaches like QMIX, VDN, or MADDPG. This choice of a relatively simple baseline algorithm makes it difficult to assess the true value of the proposed LLM-mediated guidance system. Furthermore, the evaluation is conducted exclusively on a new Aerial Wildfire Suppression environment, raising questions about the method's generalizability to established MARL benchmarks like StarCraft II or Hanabi.

The experimental results themselves raised concerns. Although the authors addressed some issues through additional experiments during the rebuttal period, including new scalability tests with varying numbers of agents, the fundamental limitations in the experimental design remain. The comparison with the no-intervention baseline may not be entirely fair, as pointed out by Reviewer W4b8, since the baseline agents lack access to certain global information available to the intervention-based approaches.

Several reviewers also noted technical concerns about the integration of LLM interventions with the PPO training process. The paper lacks discussion of how the intervention-driven actions affect the on-policy nature of PPO training, particularly regarding the handling of importance sampling for off-policy data. This raises questions about the mathematical validity of the proposed approach.

While two reviewers increased their scores to 6 after the authors' responses and additional experiments, three reviewers maintained their rejection recommendations, citing continuing concerns about technical depth, experimental rigor, and comparative evaluation. The divergence in reviewer opinions reflects the paper's mixed contributions - while the core idea is interesting, the execution and validation need substantial strengthening.

For future submissions, I recommend the authors focus on (1) expanding the evaluation to include comparisons with state-of-the-art CTDE-based MARL methods, (2) testing the approach on established MARL benchmarks, and (3) providing deeper technical analysis of the interaction between LLM interventions and MARL training.

**Additional Comments On Reviewer Discussion:**

During the rebuttal period, there was extensive discussion between the authors and reviewers about several key aspects of the paper. The main points of concern raised by reviewers centered on experimental validation, technical depth, and comparative evaluation.

Reviewer W4b8 raised concerns about the experimental results, particularly regarding the large uncertainties in training curves and potential unfairness in baseline comparisons. The authors responded by identifying and fixing a bug in their action-framerates synchronization, leading to new results showing more significant improvements in performance metrics. While these updated results demonstrated clearer performance gains, some reviewers remained unconvinced about the fundamental comparison fairness.

Reviewers XdBm and 5BcY questioned the novelty and technical depth of the contribution. In response, the authors added more detailed pseudocode explaining their MARL training process and clarified their use of CTDE. However, the lack of comparison with state-of-the-art MARL approaches remained a significant limitation.

Reviewer rf94 and eHD7 provided constructive feedback focusing on the need for more comprehensive analysis and scalability testing. The authors responded positively to these suggestions, conducting additional experiments with varying numbers of agents (3-6) and providing more detailed results in the appendix. This response led these reviewers to increase their scores from 5 to 6, acknowledging the improved empirical validation.

However, three reviewers (XdBm, 5BcY, and W4b8) maintained their rejection recommendations even after the authors' responses. Their persistent concerns about limited technical novelty, inadequate comparative evaluation, and methodological issues significantly influenced the final decision. While the authors made commendable efforts to address reviewer feedback through additional experiments and clarifications, the core limitations in experimental design and technical depth remained unresolved.

The divergent reviewer responses - with some increasing their scores while others maintained rejection recommendations - highlighted the paper's mixed contributions. This split in reviewer opinion, combined with the remaining technical and experimental limitations, supported the decision to reject the submission.

---

### Decision · Program_Chairs · 2025-01-22

Reject